# FAIRNESS UNDER DEMOGRAPHIC SCARCE REGIME

## ABSTRACT

Most existing works on fairness assume the model has full access to demographic information. However, there exist scenarios where demographic information is partially available because a record was not maintained throughout data collection or due to privacy reasons. This setting is known as *demographic scarce regime*. Prior research have shown that training an attribute classifier to replace the missing sensitive attributes (*proxy*) can still improve fairness. However, the use of proxy-sensitive attributes worsens fairness-accuracy trade-offs compared to true sensitive attributes. To address this limitation, we propose a framework to build attribute classifiers that achieve better fairness-accuracy trade-offs. Our method introduces uncertainty awareness in the attribute classifier and enforces fairness on samples with demographic information inferred with the lowest uncertainty. We show empirically that enforcing fairness constraints on samples with uncertain sensitive attributes is detrimental to fairness and accuracy. Our experiments on five datasets showed that the proposed framework yields models with significantly better fairness-accuracy trade-offs compared to classic attribute classifiers. Surprisingly, our framework can outperform models trained with constraints on the true sensitive attributes in most benchmarks.

## 1 INTRODUCTION

Mitigating machine learning bias against certain demographic groups becomes challenging when demographic information is wholly or partially missing. Demographic information can be missing for various reasons, e.g. due to legal restrictions, prohibiting the collection of sensitive information of individuals, or due to the disclosure of such information being voluntary. As people are more concerned about their privacy, reluctant users will not provide their sensitive information. As such, demographic information is available only for a few users. A *demographic scarce* regime was the term used by Awasthi et al. (2021) to describe this particular setting. The data in this setting can be divided into two different sets $\mathcal{D}_1$ and $\mathcal{D}_2$. The dataset $\mathcal{D}_1$ does not contain demographic information while $\mathcal{D}_2$ contains both sensitive and non-sensitive information. The goal is to train a classifier that is fair with respect to different (unobserved) demographic groups in $\mathcal{D}_1$. Without demographic information in $\mathcal{D}_1$, it is more challenging to enforce group fairness notions such as *statistical parity* (Dwork et al., 2012) and *equalized odds* (Hardt et al., 2016). Algorithms to enforce these notions require access to sensitive attributes in order to quantify and mitigate the model's disparities across different groups (Hardt et al., 2016; Agarwal et al., 2018; Kenfack et al., 2021). However, having access to another dataset where sensitive attributes are available gives room to train a sensitive attribute classifier that can serve as a *proxy* for the missing ones. We are interested in understanding what level of fairness/accuracy one can achieve if proxy-sensitive attributes are used in replacement of the true sensitive attributes as well as properties of the sensitive attribute classifier and the data distribution that influence the fairness-accuracy trade-off.

In their study, Awasthi et al. (2021) demonstrated a counter-intuitive finding: when using proxy-sensitive attributes, neither the highest accuracy nor an equal error rate of the sensitive attribute classifier has an impact on the accuracy of the bias estimation. Although Gupta et al. (2018) showed that improving fairness for the *proxy* demographic group can improve fairness with respect to the true demographic group, it remains unclear how existing fairness mechanisms would perform in the presence of proxy-sensitive attributes and what fairness-accuracy level they can achieve compared to the use of actual sensitive attributes when the latter is not available. *What is the optimal way for practitioners to design sensitive attribute classifiers and integrate them into existing fairness-enhancing methods in a way to achieve a better trade-off between accuracy and fairness? How does*

*sensitive attribute imputation impact fairness-accuracy tradeoffs when demographic information is missing?* We aim to answer these questions and provide insights into the characteristics of the data distribution and the attribute classifier that can yield better performances in terms of fairness and accuracy.

**Hypothesis.** We hypothesize that samples with *reliable* demographic information should be used to fit fairness constraints, backed up by the intuition that these samples are *easier* to discriminate against, while samples with uncertain demographic information are already hard to discriminate against.

In this paper, we show that existing fairness-enhancing methods are robust to noise introduced in the sensitive attribute space by the proxy attribute classifier, i.e., there is no significant gap between fairness-accuracy trade-off achieved by fairness algorithms considered when proxy attributes are used in replacement to the sensitive attribute. We hypothesize that the uncertainty of the sensitive attribute classifier plays a critical role in improving fairness-accuracy tradeoffs on downstream tasks. We show that samples whose sensitive attribute values are predicted by the attribute classifier with high uncertainty are *detrimental* to the fairness-accuracy trade-off on downstream tasks. As such, we show empirically that existing fairness-enhancing methods achieve better fairness-accuracy trade-offs when fairness constraints are enforced only on samples whose sensitive attribute values are predicted with low uncertainty. To validate our hypothesis, we propose a framework that consists of two phases. During the first phases, we construct an uncertainty-aware Deep Neural Network (DNN) model to predict demographic information. With semi-supervised training, our method measured the uncertainty and improved it during the training using Monte Carlo dropout (Gal & Ghahramani, 2016). The first phase outputs for each data point, the predicted sensitive attribute, and the uncertainty of the prediction. During the second phase, the classifier for the target task is trained with fairness constrained w.r.t to predicted sensitive attributes. However, fairness constraints are imposed only on samples whose sensitive attribute values are predicted with low uncertainty. Our main contributions are summarized as follows:

- We show that data imputation can be a good strategy for handling missing demographic information , i.e., when the sensitive attribute is missing for some samples, replacing them using imputation techniques based on the nearest neighbor or DNN models can still yield a reasonably fair model. However, the fairness-accuracy tradeoff achieved is suboptimal compared to the model trained with the true sensitive attributes.

- We propose a framework that demonstrates that accounting for the uncertainty of sensitive attribute predictions can play an important role in achieving better accuracy-fairness trade-offs. We hypothesize that a better fairness-accuracy trade-off can be achieved when fairness constraints are imposed on samples whose sensitive attribute values are predicted with high confidence by a DNN. We also show that a model trained without fairness constraints but using data with high uncertainty in the predictions of sensitive attributes tends to be fairer.

- We perform experiments on a wide range of real-world datasets to demonstrate the effectiveness of the proposed framework compared to existing methods. In essence, our results also show that the proposed method can significantly outperform a model trained with fairness constraints on observed sensitive attributes. This suggests that applying our method in settings where demographic information is fully available can yield better fairness-accuracy trade-offs.

## 2 RELATED WORK.

Various metrics have been proposed in the literature to measure unfairness in classification, as well as numerous methods to enforce fairness as per these metrics. The most popular fairness metrics include demographic parity (Dwork et al., 2012), equalized odds, and equal opportunity (Hardt et al., 2016). Demographic parity enforces the models' positive outcome to be independent of the sensitive attributes, while equalized odds aim at equalizing models' true positive and false positive rates across different demographic groups. Fairness-enhancing methods are categorized into three groups: pre-processing (Zemel et al., 2013; Kamiran & Calders, 2012), in-processing (Agarwal et al., 2018; Zhang et al., 2018), and post-processing (Hardt et al., 2016), depending on whether the fairness constraint is enforced before, during, or after model training respectively. However, enforcing these fairness notions often requires access to demographic information. There are fairness

notions that do not require demographic information to be achieved, such as the *Rawlsian Max-Min* fairness notion (Rawls, 2020) which aims at maximizing the utility of the worst-case (unknown) group (Hashimoto et al., 2018; Lahoti et al., 2020; Liu et al., 2021; Levy et al., 2020). Specifically, these methods focus on maximizing the accuracy of the unknown worst-case group. However, they often fall short in effectively targeting the specific disadvantaged demographic groups or improving group fairness metrics (Franke, 2021; Lahoti et al., 2020). In contrast, we are interested in achieving group fairness notions via proxy using limited demographic information. Recent efforts have explored bias mitigation when demographic information is noisy (Wang et al., 2020; Chen et al., 2022a). Noise can be introduced in the sensitive feature space due to human annotation, privacy mechanisms, or inference (Chen et al., 2022b). Chen et al. (2022a) aims to correct the noise in the sensitive attribute space before using them in fairness-enhancing algorithms. Another line of work focuses on alleviating privacy issues in collecting and using sensitive attributes. This group of methods aims to train fair models under privacy-preservation of the sensitive attributes. They design fair models using privacy-preserving mechanisms such as trusted third party (Veale & Binns, 2017), secure multiparty computation (Kilbertus et al., 2018), and differential privacy (Jagielski et al., 2019).

The most related work includes methods relying on proxy-sensitive attributes to enforce fairness when demographic information is partially available. Coston et al. (2019); Liang et al. (2023) assumed sensitive attribute is available either in a source domain or the target domain, and used domain adaptation-like techniques to enforce fairness in the domain with missing sensitive attributes. Diana et al. (2022) showed that training a model to predict the sensitive attributes can serve as a good substitute for the ground truth sensitive attributes when the latter is missing. Awasthi et al. (2021) showed that one can leverage samples with sensitive attribute values to create a sensitive attribute predictor that can then infer the missing sensitive attribute values. They then proposed an active sampling approach to improve bias assessment when predicted sensitive attributes are used. Gupta et al. (2018) used non-protected features to infer proxy demographic information in replacement to the unobserved real ones. They showed empirically that enforcing fairness with respect to proxy groups generalizes well to the real protected groups and can be effective in practice. While they focus on post-processing techniques, we are interested in in-processing methods.

Related work relying on proxy-sensitive attributes mostly focuses on assessing what level of fairness can be achieved when proxy-sensitive attributes are used (Coston et al., 2019), properties of the sensitive attribute classifier (Diana et al., 2022; Coston et al., 2019), and bias assessment via proxy sensitive features (Awasthi et al., 2021). Our proposed method focuses on reducing accuracy-fairness trade-offs yield by models using proxy attributes in replacement to true sensitive attributes.

## 3  PROBLEM SETTING AND PRELIMINARIES.

**Problem formulation.** We consider a dataset $\mathcal{D}_1 = \{\mathcal{X}, \mathcal{Y}\}$ where $\mathcal{X} = \{x_i\}_{i=1}^{M}$ represents the non-sensitive input feature space and $\mathcal{Y} = \{0, 1\}$ represents the target variable. The goal is to build a classifier, $f : \mathcal{X} \rightarrow \mathcal{Y}$, that can predict $\mathcal{Y}$ while ensuring fair outcomes for samples from different demographic groups. However, demographic information of samples in $\mathcal{D}_1$ is unknown. We assume the existance of another dataset $\mathcal{D}_2 = \{\mathcal{X}, \mathcal{A}\}$ sharing the same input feature space as $\mathcal{D}_1$ and for which demographic information is available, i.e., $\mathcal{A} = \{0, 1\}$. We assume binary demographic groups for simplicity. Therefore, the dataset $\mathcal{D}_1$ contains label information and $\mathcal{D}_2$ contains demographic information. Our goal is to leverage $\mathcal{D}_2$ to train an attribute classifier $g : \mathcal{X} \rightarrow \mathcal{A}$ that can serve as a proxy to the sensitive attributes for samples in $\mathcal{D}_1$, for which a fairness metric can be enforced in a way to improve fairness with respect to the true sensitive attributes. Attribute classifiers have been used in health (Brown et al., 2016; Fremont et al., 2005) and finance (Zhang, 2018; Silva et al., 2019) to infer missing sensitive attributes, in particular when users or patients self-report their protected information. To be able to estimate the true disparities in the label classifier $f$, we assume there exists a small set of samples drawn from the joint distribution $\mathcal{X} \times \mathcal{Y} \times \mathcal{A}$, i.e., samples that jointly have label and demographic information. If this subset is not available, one can consider using the active sampling technique proposed by Awasthi et al. (2021) in order to approximate bias with respect to the predicted sensitive attributes. This estimation is beyond the scope of this work. Our goal is to effectively assess the level of fairness our method can achieve without being overly concerned about potential bias overestimation or underestimation. Reducing the trade-off between fairness and accuracy is a significant challenge within the fair machine-learning community (Dutta et al., 2020). Our primary goal is to design a method that effectively leverages proxy features to achieve

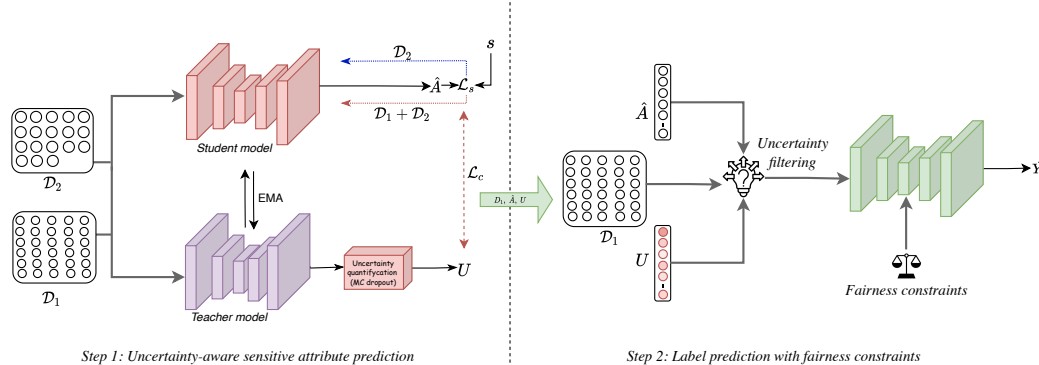

Figure 1: Overview of the proposed method. Our framework consists of two steps. In the first step (left), the dataset $\mathcal{D}_2$ is used to train the attribute classifier for the student-teacher framework. The first step produces proxy-sensitive attributes ($g(X) = \hat{A}$) and the uncertainty of their predictions ($U$). In the second step (right), only samples with reliable proxy-sensitive attributes are used to train the fair model. These samples are selected based on a defined threshold of their uncertainties.

similar or better fairness-accuracy trade-offs compared to settings where the true sensitive attributes are available. To this end, we considered a different range of fairness metrics along with various (in-processing) fairness-enhancing techniques.

**Fairness Metrics.** In this work, we consider three popular group fairness metrics: demographic parity (Dwork et al., 2012), equalized odds, and equal opportunity (Hardt et al., 2016). These metrics aim to equalize the model's performance across different demographic groups, see Appendix B for more details.

**Fairness Mechanism.** We focus on in-processing techniques to improve the models' fairness. These methods introduce constraints in the classification problem to satisfy a given fairness metric. Our study focuses on state-of-the-art techniques in this category, i.e., exponentiated gradient (Agarwal et al., 2018) and adversarial debiasing (Zhang et al., 2018). We considered these methods as they allow better control over fairness and accuracy. In general, the optimization problem contains a parameter $\lambda$ that controls the balance between fairness and accuracy, i.e., a higher value of $\lambda$ would force the model to achieve higher fairness (respectively lower accuracy) while a smaller yields higher accuracy (respectively lower fairness). Our goal is to design a sensitive attribute predictor that achieves a better fairness-accuracy trade-off, i.e., for the same value of $\lambda$ build a model that provides higher accuracy and lower unfairness compared to other baselines.

## 4 METHODOLOGY.

In this section, we present the methodology and all components involved in our proposed method. Figure 1 presents an overview of the stages in our framework and the interactions between and within each stage. The first stage consists of *training the attribute classifier* and outputs for each sample with missing sensitive attribute, its predicted sensitive attribute (proxy) along with the uncertainty of the prediction. In the second step, *the label classifier is trained with fairness constrained enforced using the predicted sensitive attributes*. Fairness is enforced only on samples with the lowest uncertainty in the sensitive attribute prediction, i.e., samples with an uncertainty lower than a predefined uncertainty threshold $H$.

### 4.1 UNCERTAINTY-AWARE ATTRIBUTE PREDICTION.

We build the sensitive attribute classifier using a student-teacher framework in a semi-supervised learning approach similar to (Yu et al., 2019; Laine & Aila, 2017), which accounts for the uncertainty of the predictions of samples with missing sensitive attributes.

**Student model.** The student model is implemented as a neural network and is trained on $\mathcal{D}_2$ (samples with sensitive attributes) to predict sensitive attributes. The attribute classifier is optimized to minimize a double loss function: the *classification loss* ($\mathcal{L}_s$), i.e., the cross-entropy loss, and the *consistency loss* ($\mathcal{L}_c$) (Yu et al., 2019). The consistency loss (or unsupervised loss) enforces the student model to rely mostly on samples with confident sensitive attributes guided by the uncertainty estimation from the teacher model. This loss is defined as the mean squared difference between the outputs (logits) of the student and the teacher on samples for which the uncertainty does not exceed a predefined threshold $R$. The motivation behind the consistency loss is the focus on the primary goal of the attribute classifier, which is to find the missing sensitive attributes in $\mathcal{D}_1$ with high confidence.

Overall, the attribute classifier is trained to minimize the following loss:

$$\min_{f \in \mathcal{F}} \mathbb{E}_{(x,a) \sim \mathcal{D}_2 \times \mathcal{A}} \mathcal{L}_s(f(x), a) + \lambda \mathbb{E}_{x \sim \mathcal{D}_1 + \mathcal{D}_2} \mathcal{L}_c(f(x), h(x)) \tag{1}$$

where $f(\cdot)$ is the student model, $h(\cdot)$ the teacher model, and $\lambda$ a parameter controlling the consistency loss. The empirical loss minimized is defined by the following equations for classification ($\mathcal{L}_s$) and consistency loss $\mathcal{L}_c$:

$$\mathcal{L}_s = \frac{1}{|\mathcal{D}_2|} \sum_{x,a \in \mathcal{D}_2, A} a \cdot \log(f(x)) + (1 - a) \cdot \log(1 - f(x)) \tag{2}$$

$$\mathcal{L}_c = \frac{1}{|\mathcal{D}_2| + |\mathcal{D}_1|} \sum_{x|u_x \leq R} \| f(x) - h(x) \|^2 \tag{3}$$

The consistency loss is applied only on samples, $x$, whose uncertainty, $u_x$, is lower than the predefined threshold $R$. Following Srivastava et al. (2014); Baldi & Sadowski (2013), $R$ and $\lambda$ are updated using a Gaussian warmup function to prevent the model from diverging at the beginning of the training.

**Teacher model.** The teacher model is implemented using the same network architecture as the student, and is used for uncertainty estimation. The teacher weights are updated within each training epoch, $t$, using the exponential moving average (EMA) of student weights:

$$\omega_t = \alpha \omega_{t-1} + (1 - \alpha)\theta, \tag{4}$$

where $\theta$ and $\omega$ denote the respective weights of student and teacher and $\alpha$ controls the moving decay. The use of EMA to update the teacher model is motivated by previous studies (Laine & Aila, 2017; Yu et al., 2019) that have shown that averaging model parameters at different training epochs can provide better predictions than using the most recent model weights in the last epoch. The teacher model gets as input both samples from both $\mathcal{D}_1$ and $\mathcal{D}_2$, and computes the uncertainty of their predictions using Monte Carlo (MC) dropout (Gal & Ghahramani, 2016). As such, both the student and teacher networks have dropout layers between hidden layers of the network.

MC dropout is an approximation of a Bayesian neural network widely used to interpret the parameters of neural networks (Abdar et al., 2021). It uses dropout at test time in order to compute prediction uncertainty from different sub-networks that can be derived from the whole original neural network. Dropout is generally used to improve the generalization of DNNs. During training, the dropout layer randomly removes a unit with probability $p$. Therefore, each forward and backpropagation pass is done on a different model (sub-network) forming an ensemble of models that are aggregated together to form a final model with lower variance (Srivastava et al., 2014; Baldi & Sadowski, 2013). The uncertainty of each sample is computed using $T$ stochastic forward passes on the teacher model to output $T$ independent and identically distributed predictions, i.e., $\{h_1(x), h_2(x), \cdots, h_T(x)\}$. The softmax probability of the output set is calculated and the uncertainty of the prediction ($u_x$) is quantified using the resulting entropy: $u_x = -\sum_a p_a(x) \log(p_a(x))$, where $p_a(x)$ is the probability that sample $x$ belongs to demographic group $a$ estimated over $T$ stochastic forward passes, i.e., $p_a(x) = \frac{1}{T} \sum_{t=1}^{T} h_t^a(x)$.

### 4.2 ENFORCING FAIRNESS W.R.T RELIABLE PROXY SENSITIVE ATTRIBUTES.

After the first phase, the attribute classifier can produce for every sample in $\mathcal{D}_1$, i.e., samples with missing sensitive attributes, their predicted sensitive attribute (proxy) $\hat{A} = \{h(x_i)_{x_i \in \mathcal{D}_1}\}$, and the

uncertainty of the prediction $U = \{u_{x_i}\}_{x_i \in \mathcal{D}_1}$. To validate our hypothesis, we define a confidence threshold $H$ for samples used to train the label classifier with fairness constraints, i.e., the label classifier with fairness constraints is trained on a subset $\mathcal{D}'_1 \subset \mathcal{D}_1$ defined as follows:

$$\mathcal{D}'_1 = \{(x, y, f(x)) | u_x \leq H\} \tag{5}$$

The hypothesis of enforcing fairness on samples whose sensitive attributes are reliably predicted stems from the fact that the model is confidently able to distinguish these samples based on their sensitive attributes in the latent space. In contrast, the label classifier is inherently fairer if an attribute classifier cannot reliably predict sensitive attributes from training data (Kenfack et al., 2023). We further support this in section 5.2 by comparing the new Adult dataset (Ding et al., 2021) and the old version of the dataset (Asuncion & Newman, 2007). Therefore, enforcing fairness constraints on samples with the most reliable proxy attributes would be more useful in achieving better accuracy-fairness trade-offs than considering samples for which the sensitive attributes are not distinguishable in the latent space. The fairness constraints on samples with unreliable sensitive attributes could push the model's decision boundary in ways that penalize accuracy and/or fairness. We support these arguments in the experiments.

## 5 EXPERIMENTS

In this section, we demonstrate the effectiveness of our framework on five datasets and compare it to different baselines. The source code for reproducibility is available at https://anonymous.4open.science/r/source-code-E86F.

### 5.1 EXPERIMENTAL SETUP

**Datasets.** We validate our method on five real-world benchmarks widely used for bias assessment: Adult Income (Asuncion & Newman, 2007)[1], Compas (Jeff et al., 2016), Law school (LSAC), CelebA (Liu et al., 2018) (Wightman, 1998), and the New Adult Ding et al. (2021) dataset. More details about the datasets appear in Supplementary C.

**Attribute classifier.** The student and teacher models were implemented as feed-forward Multi-layer Perceptrons (MLPs) with Pytorch (Paszke et al., 2019), and the loss function 1 is minimized using the Adam optimizer (Kingma & Ba, 2014) with learning rate $0.001$ and batch size $256$. Following Yu et al. (2019); Laine & Aila (2017), we used $\alpha = 0.99$ for the EMA parameter for updating the teacher weights using the student's weights across epochs. The uncertainty threshold is finetuned over the interval $[0.1, 0.7]$ using 10% of the training data. The best-performing threshold is used for the thresholding in the second step to obtain $\mathcal{D}'_1$. The uncertainty threshold that achieved the best results are $0.30, 0.60, 0.66, 0.45$ for the Adult, Compas, LSAC, and CelebA datasets, respectively.

**Baselines.** For fairness-enhancing mechanisms, we considered the Fairlean (Bird et al., 2020) implementation of the exponentiated gradient (Agarwal et al., 2018). We considered two variants of our approach: a variant where the model is trained without fairness constraints but using samples with higher uncertainty in the sensitive attribute predictions — *Ours (uncertain)*, and a variant where only samples with reliable (certain) attributes are used to train the label classifier with fairness constraints using the exponentiated gradient — *Ours (certain)*. For comparison, we considered methods that aim to improve fairness without (full) demographic information. We compare with the following methods:

- FairFS (Zhao et al., 2022): This method assumes that non-sensitive features that correlate with sensitive attributes are known. It leverages these related features to improve fairness w.r.t the unknown sensitive attributes.
- FairDA (Liang et al., 2023): Similar to our setting, this method assumes the sensitive information is available in a *source domain* (dataset $D_2$ in our setting). It uses a domain adaptation-based approach to transfer demographic information from the source domain to improve fairness in the target using an adversarial approach.

---

[1]https://archive.ics.uci.edu/ml/datasets/Adult

| Method | Accuracy | $\Delta_{DP}$ | $\Delta_{EOP}$ | $\Delta_{EOD}$ |
|---|---|---|---|---|
| Vanilla (without fairness) | $0.851 \pm 0.008$ | $0.171 \pm 0.004$ | $0.088 \pm 0.033$ | $0.091 \pm 0.030$ |
| Vanilla (with fairness) | $0.829 \pm 0.002$ | $0.005 \pm 0.004$ | $0.021 \pm 0.014$ | $0.017 \pm 0.007$ |
| FairRF | $0.838 \pm 0.002$ | $0.162 \pm 0.015$ | $0.063 \pm 0.027$ | $0.072 \pm 0.019$ |
| FairDA | $0.809 \pm 0.009$ | $0.087 \pm 0.028$ | $0.071 \pm 0.046$ | $0.078 \pm 0.039$ |
| ARL | $\mathbf{0.850} \pm 0.002$ | $0.173 \pm 0.013$ | $0.028 \pm 0.09$ | $0.097 \pm 0.031$ |
| CVarDRO | $0.820 \pm 0.012$ | $0.200 \pm 0.005$ | $0.160 \pm 0.03$ | $0.100 \pm 0.027$ |
| KSMOTE | $0.814 \pm 0.003$ | $0.302 \pm 0.007$ | $0.160 \pm 0.021$ | $0.196 \pm 0.003$ |
| DRO | $0.823 \pm 0.003$ | $0.184 \pm 0.042$ | $0.092 \pm 0.041$ | $0.105 \pm 0.041$ |
| Ours (uncertain) | $0.825 \pm 0.013$ | $0.106 \pm 0.036$ | $0.065 \pm 0.047$ | $0.068 \pm 0.032$ |
| Ours (certain) | $0.830 \pm 0.004$ | $\mathbf{0.007} \pm 0.005$ | $\mathbf{0.015} \pm 0.010$ | $\mathbf{0.018} \pm 0.016$ |

Table 1: Comparison with different baselines on the Adult dataset. Bolded values represent the best-performing baselines among the fairness-enhancing methods without (full) demographic information.

- ARL (Lahoti et al., 2020): The method uses an adversarial approach to upweight samples in regions hard to learn, i.e., regions where the model makes the most mistakes.
- Distributionally Robust Optimization (DRO) (Hashimoto et al., 2018): It optimizes for the worst-case distribution around the empirical distribution. Similar to ARL, the goal is to improve the accuracy of the worst-case group.
- CVarDRO (Levy et al., 2020): It is an improved variant of DRO.
- KSMOKE (Yan et al., 2020) performs clustering to obtain pseudo groups and use them as substitutes to oversample the minority groups.

For each baseline, we used the code provided by the authors[2] along with the recommended hyperparameters. We considered the case where the sensitive attribute is fully available and trained the model with fairness constraints w.r.t the ground truth (Vanilla (with fairness)) using exponentiated gradient (Agarwal et al., 2018). For comparison, in addition to the accuracy, we consider the three fairness metrics described in the appendix B, i.e., equalized odds ($\Delta_{EOD}$), equal opportunity ($\Delta_{EOP}$), and demographic parity ($\Delta_{DP}$). All the baselines are trained on 70% of the $\mathcal{D}_1$, and fairness and accuracy are evaluated on the 30% as the test set. To report the true fairness violation, we assume the sensitive attribute is observed in the test set. We trained each baseline 7 times and averaged the results. We use logistic regression[3] as the base classifier for all the baselines and train each baseline to achieve maximum fairness.

## 5.2 Results and Discussion

**Uncertainty of the sensitive attribute and fairness.** Table 4 showcases the average uncertainty of the sensitive attribute prediction estimated by our method. The table also shows different fairness measures of a logistic regression model trained without fairness constraints on the dataset $\mathcal{D}_1$. We observe that the uncertainty in the Adult dataset is lower compared to the New adult while the unfairness in the Adult dataset is higher. These results show the correlation between the uncertainty of the sensitive attribute prediction and the fairness of the model. In particular, the least biased dataset (LSAC) has the highest uncertainty of the sensitive attribute (0.66) while for datasets with lower uncertainty, the unfairness is higher, e.g., the Adult and CelebA datasets. This provides evidence to support our hypothesis that a model can hardly discriminate against samples with uncertain demographic groups. Furthermore, we show that if we train a model without fairness constraints, but using samples with high uncertainty in the prediction of their sensitive attributes, the fairness of the predictions can be improved (See Supplementary D).

**Fairness-accuracy trade-offs.** Table 1, 2, and 3 show the effectiveness of the proposed method compared to other baselines on the Adult, Compas, and LSAC datasets respectively ( results for CelebA appear in Appendix, Table 6). It is important to note that methods that aim to achieve worst-case groups (ARL, DRO, CVarDRO) do not necessarily improve fairness in terms of demographic

---

[2]We implemented FairDA and reproduced using the instructions in the paper (Liang et al., 2023).

[3]Appendix E shows a comparison with an MLP model. )

| Method | Accuracy | $\Delta_{DP}$ | $\Delta_{EOP}$ | $\Delta_{EOD}$ |
|---|---|---|---|---|
| Vanilla (without fairness) | $0.681 \pm 0.011$ | $0.285 \pm 0.026$ | $0.325 \pm 0.029$ | $0.325 \pm 0.029$ |
| Vanilla (with fairness) | $0.634 \pm 0.009$ | $0.032 \pm 0.011$ | $0.039 \pm 0.024$ | $0.041 \pm 0.016$ |
| FairRF | $0.669 \pm 0.001$ | $0.289 \pm 0.003$ | $0.319 \pm 0.004$ | $0.319 \pm 0.004$ |
| FairDA | $0.668 \pm 0.019$ | $0.229 \pm 0.018$ | $0.265 \pm 0.024$ | $0.265 \pm 0.024$ |
| ARL | $0.672 \pm 0.009$ | $0.290 \pm 0.016$ | $0.310 \pm 0.010$ | $0.320 \pm 0.010$ |
| CVarDRO | $0.668 \pm 0.008$ | $0.279 \pm 0.018$ | $0.300 \pm 0.010$ | $0.287 \pm 0.015$ |
| KSMOTE | $0.670 \pm 0.012$ | $0.286 \pm 0.028$ | $0.321 \pm 0.028$ | $0.321 \pm 0.028$ |
| DRO | $0.672 \pm 0.010$ | $0.282 \pm 0.026$ | $0.296 \pm 0.017$ | $0.296 \pm 0.017$ |
| Ours (uncertain) | $0.671 \pm 0.009$ | $0.272 \pm 0.016$ | $0.300 \pm 0.039$ | $0.300 \pm 0.034$ |
| Ours (certain) | $\mathbf{0.676} \pm 0.009$ | $\mathbf{0.085} \pm 0.016$ | $\mathbf{0.067} \pm 0.039$ | $\mathbf{0.074} \pm 0.034$ |

Table 2: Comparison with different baselines on the Compas dataset.

| Method | Accuracy | $\Delta_{DP}$ | $\Delta_{EOP}$ | $\Delta_{EOD}$ |
|---|---|---|---|---|
| Vanilla (without fairness) | $0.793 \pm 0.007$ | $0.014 \pm 0.005$ | $0.005 \pm 0.005$ | $0.049 \pm 0.026$ |
| Vanilla (with fairness) | $0.796 \pm 0.009$ | $0.004 \pm 0.004$ | $0.002 \pm 0.001$ | $0.025 \pm 0.016$ |
| FairRF | $0.753 \pm 0.120$ | $0.021 \pm 0.013$ | $0.016 \pm 0.017$ | $0.044 \pm 0.015$ |
| FairDA | $0.716 \pm 0.210$ | $0.001 \pm 0.000$ | $0.000 \pm 0.005$ | $0.003 \pm 0.004$ |
| ARL | $\mathbf{0.807} \pm 0.024$ | $0.014 \pm 0.015$ | $0.009 \pm 0.014$ | $0.037 \pm 0.022$ |
| CVarDRO | $0.776 \pm 0.052$ | $0.024 \pm 0.010$ | $0.019 \pm 0.014$ | $0.045 \pm 0.015$ |
| KSMOTE | $0.655 \pm 0.055$ | $0.022 \pm 0.034$ | $0.030 \pm 0.022$ | $0.060 \pm 0.018$ |
| DRO | $0.580 \pm 0.220$ | $0.023 \pm 0.014$ | $0.021 \pm 0.017$ | $0.038 \pm 0.020$ |
| Ours (uncertain) | $0.794 \pm 0.001$ | $0.015 \pm 0.002$ | $0.006 \pm 0.001$ | $0.055 \pm 0.000$ |
| Ours (certain) | $0.805 \pm 0.001$ | $\mathbf{0.001} \pm 0.002$ | $\mathbf{0.000} \pm 0.001$ | $\mathbf{0.002} \pm 0.000$ |

Table 3: Comparison with different baselines on the LSAC dataset.

party or equalized odds. In particular, Tables 1, 2, 3 show that ARL can improve the Equal Opportunity metric but fails to improve demographic parity. It also yields the most accurate classifier as this method does not have a tradeoff with accuracy. On the other hand, FairDA, which also exploits limited demographic information, shows an improvement in fairness compared to other baselines. However, it incurs a higher drop in accuracy while our method using reliable sensitive attributes outperforms it across all datasets. Overall, the results show that our method with fairness constraints on samples with reliable sensitive attributes provides Pareto dominant points in terms of fairness and accuracy. On the other hand, the variant using a model trained without fairness constraints (without using sensitive attributes) provides better fairness-accuracy trade-offs compared to other baselines on the Adult and the CelebA datasets while providing comparable results on datasets with higher uncertainty (LSAC and Compas). For example, the LSAC dataset has an average uncertainty of 0.66, meaning most samples already have uncertain sensitive information and the unfairness is already low. As no fairness constraints are enforced in *Ours (uncertain)*, it has less impact on fairness and accuracy when most data samples are preserved due to high overall uncertainty.

**Impact of the uncertainty threshold.** Figure 2 showcases the impact of the uncertainty threshold on the fairness-accuracy threshold. When the feature space encodes much information about the sensitive attribute as in the Adult dataset (Figure 2a) with 85% accuracy of predicting the sensitive attributes, results show that the more we enforce fairness w.r.t. samples with the lowest uncertainty, the better the fairness-accuracy trade-offs. In this regime, enforcing unfairness helps the model maintain a better accuracy level (Figure 2a). In contrast, in a low bias regime, i.e. when the feature space does not encode enough information about the sensitive attributes such as on the Compas and the New Adult dataset, the model achieves better fairness-accuracy trade-offs when a higher uncertainty threshold is used. In this regime, most of the samples have higher uncertainty in the sensitive attribute prediction (see Table 4), as can be observed in Figure 2b and Figure 2c, the use of a lower uncertainty threshold leads to a decrease in accuracy while fairness is improved. We observe similar results in the CelebA and LSAC datasets (Fig 9 in Supplementary). The drop in accuracy is due to the fact that more and more samples were pruned out from the datasets, and this suggests that the feature space is more informative for the target task than the demographic information. In the

| Dataset | Mean uncertainty (↓) | Accuracy sensitive attribute (↑) | $\Delta_{DP}$ | $\Delta_{EOD}$ | $\Delta_{EOP}$ |
|---|---|---|---|---|---|
| Adult | 0.15 | 85% | 0.18 | 0.20 | 0.13 |
| New Adult | 0.42 | 68% | 0.06 | 0.05 | 0.04 |
| Compas | 0.39 | 72% | 0.28 | 0.32 | 0.32 |
| LSAC | 0.66 | 55% | 0.014 | 0.005 | 0.049 |
| CelebA | 0.21 | 83% | 0.17 | 0.19 | 0.19 |

Table 4: Average uncertainty and fairness of the attribute classifier on the dataset with missing sensitive attributes.

appendix ( G), we show that while under-represented demographic groups can have higher uncertainty on average than well-represented groups, minority groups are still consistently represented when a lower threshold is used.

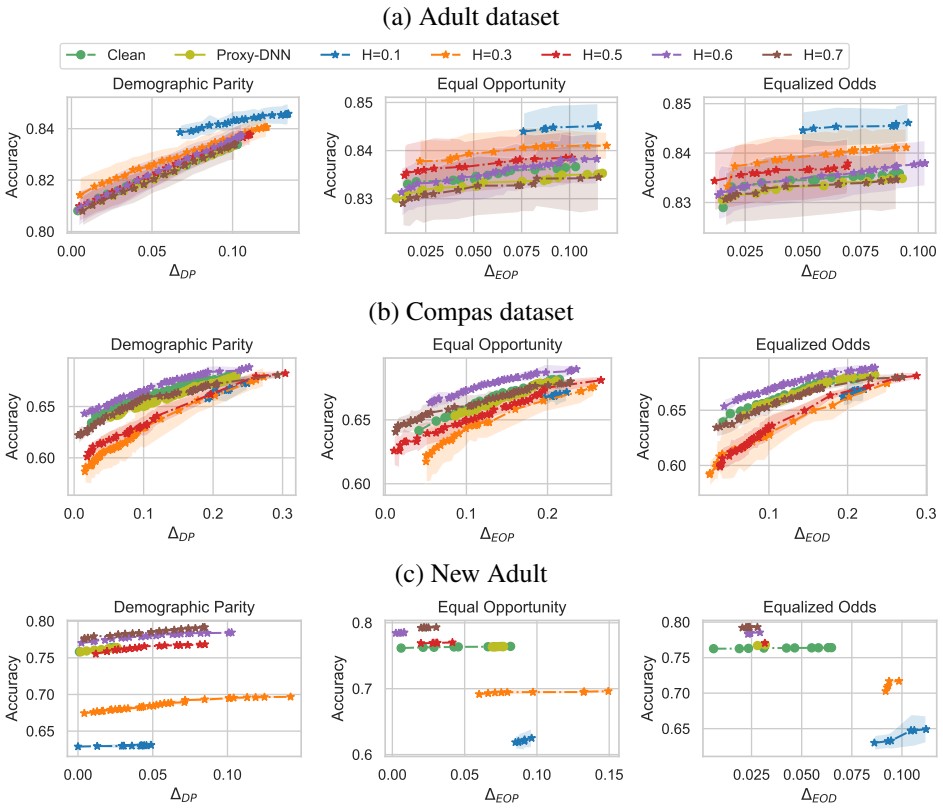

Figure 2: The impact of the uncertainty threshold $H$ on the fairness-accuracy trade-off for (a) Adult, (b) Compas, and (c) New Adult datasets.

## 6    CONCLUSION

In this work, we introduced a framework to improve the fairness-accuracy trade-off when only limited demographic information is available. Our method introduces uncertainty awareness in the sensitive attributes classifier. We showed that uncertainty in the attribute classifier plays an important role in the fairness-accuracy trade-offs achieved in the downstream model with fairness constraints. Our method consistently achieved a better trade-off than existing methods and in most cases even better trade-offs than the use of the true sensitive attributes. However, in a low-bias regime, most samples have uncertain sensitive attributes leading to a decrease in the accuracy. In future work, we plan to introduce weighted empirical risk minimization in the fairness-enhancing model where the samples' weights are defined based on the uncertainty of the attribute classifier.

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

SUPPLEMENTARY MATERIAL

## A    LIMITATIONS

While the framework proposed shows evidence that models can hardly discriminate against samples with low uncertainty in the sensitive attribute prediction, our method relies on the prediction of missing sensitive information. Inferring sensitive information can raise ethical concerns and face legal restrictions, especially when individuals do not choose to disclose their sensitive information. The line of methods relying upon proxy attributes or inferring sensitive attributes faces this limitation (Diana et al., 2022; Awasthi et al., 2021; Coston et al., 2019). Furthermore, the inference of the sensitive attributes (using our proposed method or others) should not be used for a purpose different from bias assessment mitigation. Moreover, we showed that with a variant of our method (*Ours (uncertain)*), it is possible to train a fairer model without fairness constraints but using samples with high uncertainty in the sensitive attribute predictions. This variant can operate without (predicted) sensitive information, thereby alleviating potential legal or ethical concerns associated with the prediction of sensitive information. We show in Appendix D the impact of the uncertainty threshold on the fairness of a model trained without fairness constraints. Another limitation of our method is the evaluation focuses mostly on one fairness-enhancing algorithm (i.e., Exponentiated Gradient). It will be interesting to explore if our hypothesis applies to pre-processing and post-processing techniques, and with different fairness-enhancing algorithms. Finally, our assumption that the true sensitive attributes are available in the test dataset for fairness evaluation might not be true in several practical scenarios. This might require evaluation using proxy-sensitive attributes. These proxies are likely noisy and might require evaluations using bias assessment methods that effectively quantify fairness violation w.r.t to true sensitive attribute (Chen et al., 2019; Awasthi et al., 2021).

## B    FAIRNESS METRICS

In this work, we consider three popular group fairness metrics: demographic parity, equalized odds, and equal opportunity. These metrics aim to equalize different models' performances across different demographic groups. Samples belong to the same demographic group if they share the same demographic information, $A$, e.g., gender and race.

### B.1    DEMOGRAPHIC PARITY

Also known as statistical parity, this measure requires that the positive prediction ($f(X) = 1$) of the model be the same regardless of the demographic group to which the user belongs (Dwork et al., 2012). More formally the classifier $f$ achieves demographic parity if $P(f(X) = 1|A = a) = P(f(X) = 1)$ In other words, the outcome of the model should be independent of sensitive attributes. In practice, this metric is measured as follows:

$$\Delta_{\text{DP}}(f) = \left| \mathop{\mathbb{E}}_{x|A=0}[\mathbb{I}\{f(x) = 1\}] - \mathop{\mathbb{E}}_{x|A=1}[\mathbb{I}\{f(x) = 1\}] \right| \tag{6}$$

Where $\mathbb{I}(\cdot)$ is the indicator function.

### B.2    EQUALIZED ODDS

This metric enforces the True Positive Rate (TPR) and False Positive Rate (FPR) for different demographic groups to be the same $P(f(X) = 1|A = 0, Y = y) = P(f(X) = 1|A = 1, Y = y),:$ $; \forall y \in \{0, 1\}$. The metric is measured as follows:

$$\Delta_{\text{EOD}} = \alpha_0 + \alpha_1 \tag{7}$$

Where,

$$\alpha_j = \left| \mathop{\mathbb{E}}_{x|A=0,Y=j}[\mathbb{I}\{f(x) = 1\}] - \mathop{\mathbb{E}}_{x|A=1,Y=j}[\mathbb{I}\{f(x) = 1\}] \right| \tag{8}$$

| Dataset | Size | #features | Domain | Sensitive attribute |
|---------|------|-----------|--------|---------------------|
| Adult Income | 48,842 | 15 | Finance | Gender |
| Compas | 6,000 | 11 | Criminal justice | Race |
| LSAC | 20,798 | 12 | Education | Gender |
| New Adult | 1.6M | 10 | Finance | Gender |
| CelebA | 202,600 | 40 | Image | Gender |

Table 5: Datasets

### B.3 EQUAL OPPORTUNITY

In certain situations, bias in positive outcomes can be more harmful. Therefore, Equal Opportunity metric enforces the same TPR across demographic (Hardt et al., 2016) and is measured using $\alpha_1$ (Eq. 8).

## C DATASETS

In the Adult dataset, the goal is to predict if an individual's annual income is greater or less than $50k per year. We also considered the recent version of the Adult dataset (New Adult) for the year 2018 across different states in US (Ding et al., 2021). The goal in the Compas data is to predict whether a defendant will recidivate within two years. The LSAC dataset contains admission records to law school. The task is to predict whether a candidate would pass the bar exam. The CelebA[4] dataset contains 40 facial attributes of humaine annotated images. We consider the task of predicting *attractiveness* using gender as a sensitive attribute. We randomly sample 20% of the data to train the sensitive attribute classifier ($\mathcal{D}_2$). For all the datasets, all the features are used to train the attribute classifier except for the target variable. Table 5 provides more details about each dataset and sensitive attributes used.

## D USING THE UNCERTAINTY OF SENSITIVE ATTRIBUTES TO TRAIN FAIR MODELS WITHOUT FAIRNESS CONSTRAINTS.

Results in Table 4 show that a model trained without fairness constraints tends to be fairer when the average uncertainty in predicting the sensitive attribute is high. This provides the intuition that training a model on samples with uncertain sensitive attributes can yield fairer outcomes. In this set of experiments, we trained different classifiers (Logistic Region and Random Forest) without fairness constraints but using training data with uncertain sensitive attributes. For different uncertainty thresholds $H \in \{0.0, 0.1, ..., 0.6\}$, we prune out samples whose uncertainty is lower than $H$ and train a model without fairness constraints using the remaining training data, i.e., $\{(x, y) \in D_1 | u_x \geq H\}$. Where $u_x$ is the estimated uncertainty provided by our method. In particular, when $H = 0$ all the data points are used for the training, and in other cases samples with uncertainty lower than $H$ are removed, and the model is trained on samples with more uncertain sensitive attributes. For each uncertainty threshold, we train the model seven times with different seeds and report fairness and accuracy in the testing set, which contains sensitive attributes.

Figures 3(a) and 3(b) show the fairness and accuracy of the Logistic Regression and Random Forest classifiers, respectively. In the figures, each column represents the results on each dataset, and the first and the second rows provide the plots for fairness and accuracy respectively. Across different datasets, the results show that unfairness decreases as the uncertainty threshold increases. We observe that the improvement in fairness is consistent for different fairness metrics considered, i.e., demographic parity, equal opportunity, and equalized odd. We also observe a decrease in the accuracy, which is justified by reduced dataset size and consequently the tradeoff with fairness. On the LSAC dataset, fairness and accuracy remain almost constant as the average uncertainty in predicting the

---

[4]http://mmlab.ie.cuhk.edu.hk/projects/CelebA.html

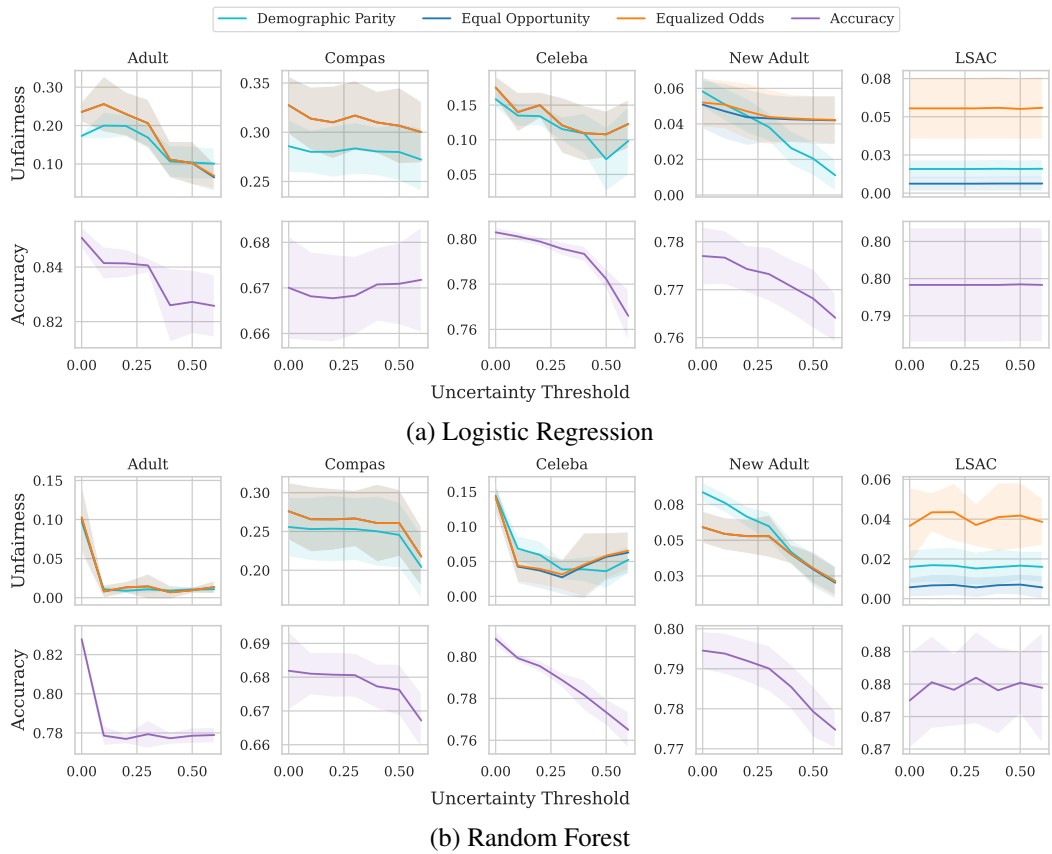

(a) Logistic Regression

(b) Random Forest

Figure 3: Training classifiers without fairness constraints using samples with high uncertainty of sensitive attribute predictions. For each uncertainty threshold $H$, the model is trained on samples with uncertainty $\geq H$. The training is done seven times and the average fairness (first row) and accuracy (second row) are reported. Shaded represents the standard deviation.

sensitive attribute on this dataset is 0.66, i.e., most of the samples already have the highest uncertainty. However, this method incurs a higher drop in accuracy and does not necessarily guarantee that an adversary cannot reconstruct the sensitive attributes from the trained model (Ferry et al., 2023).

## E    ADDITIONAL RESULTS.

Table 6 shows the comparison with other baselines on the CelebA dataset with logistic regression as the base classifier. In the main paper, we provided a comparison with other existing methods using Logistic Regression as the base classifier. We performed experiments with a more complex non-linear model in order to analyze its impact on the performance of different methods. We considered a Multi-Layer Perceptron (MLP) with one hidden layer of 32 units and with Relu as the activation function for all the baselines. On the Adult dataset, Table 7, shows that when using a more complex model, our method (Ours (certain)) still provides Pareto dominants points in terms of fairness and accuracy compared to other baselines while we observed an improvement in the accuracy of other methods due to the increased model capacity.

## F    COMPARISON WITH OTHER BASELINES

To assess the effect of the attribute classifier over the performances of downstream classifiers with fairness constraints w.r.t the proxy, we considered different methods of obtaining the missing sensitive attributes as baselines:

| Method | Accuracy | $\Delta_{\text{DP}}$ | $\Delta_{\text{EOP}}$ | $\Delta_{\text{EOD}}$ |
|---|---|---|---|---|
| Vanilla (without fairness) | $0.803 \pm 0.002$ | $0.176 \pm 0.010$ | $0.183 \pm 0.015$ | $0.183 \pm 0.015$ |
| Vanilla (with fairness) | $0.782 \pm 0.001$ | $0.008 \pm 0.005$ | $0.018 \pm 0.014$ | $0.017 \pm 0.014$ |
| FairDA | $0.802 \pm 0.002$ | $0.155 \pm 0.010$ | $0.165 \pm 0.018$ | $0.165 \pm 0.018$ |
| ARL | $\mathbf{0.803} \pm 0.002$ | $0.157 \pm 0.010$ | $0.157 \pm 0.010$ | $0.166 \pm 0.016$ |
| CVarDRO | $0.781 \pm 0.002$ | $0.155 \pm 0.010$ | $0.162 \pm 0.016$ | $0.162 \pm 0.016$ |
| KSMOTE | $0.773 \pm 0.008$ | $0.020 \pm 0.067$ | $0.110 \pm 0.082$ | $0.144 \pm 0.068$ |
| DRO | $0.796 \pm 0.006$ | $0.142 \pm 0.020$ | $0.152 \pm 0.020$ | $0.129 \pm 0.028$ |
| Ours (uncertain) | $0.782 \pm 0.004$ | $0.071 \pm 0.046$ | $0.107 \pm 0.033$ | $0.107 \pm 0.033$ |
| Ours (certain) | $0.793 \pm 0.000$ | $\mathbf{0.003} \pm 0.000$ | $\mathbf{0.001} \pm 0.001$ | $\mathbf{0.007} \pm 0.002$ |

Table 6: Comparison with different baselines on the CelebA dataset. *Ours (uncertain)* represents the variant of our approach where the model is trained without fairness constraints but using samples with higher uncertainty in the sensitive attribute predictions. And *Ours (certain)* the variant where only samples with reliable sensitive attributes are used to train the label classifier with fairness constraints using the exponentiated gradient.

| Method | Accuracy | $\Delta_{\text{DP}}$ | $\Delta_{\text{EOP}}$ | $\Delta_{\text{EOD}}$ |
|---|---|---|---|---|
| Vanilla (without fairness) | $0.853 \pm 0.004$ | $0.183 \pm 0.019$ | $0.100 \pm 0.025$ | $0.102 \pm 0.023$ |
| Vanilla (with fairness) | $0.801 \pm 0.009$ | $0.006 \pm 0.004$ | $0.049 \pm 0.011$ | $0.017 \pm 0.007$ |
| FairRF | $\mathbf{0.853} \pm 0.002$ | $0.164 \pm 0.009$ | $0.077 \pm 0.026$ | $0.091 \pm 0.013$ |
| FairDA | $0.813 \pm 0.014$ | $0.118 \pm 0.023$ | $0.091 \pm 0.050$ | $0.099 \pm 0.037$ |
| ARL | $0.851 \pm 0.003$ | $0.166 \pm 0.015$ | $0.087 \pm 0.019$ | $0.090 \pm 0.016$ |
| CVarDRO | $0.850 \pm 0.003$ | $0.183 \pm 0.018$ | $0.095 \pm 0.027$ | $0.101 \pm 0.026$ |
| KSMOTE | $0.814 \pm 0.020$ | $0.201 \pm 0.055$ | $0.120 \pm 0.021$ | $0.130 \pm 0.023$ |
| DRO | $0.837 \pm 0.016$ | $0.232 \pm 0.057$ | $0.110 \pm 0.057$ | $0.140 \pm 0.045$ |
| Ours (uncertain) | $0.801 \pm 0.027$ | $0.110 \pm 0.022$ | $0.067 \pm 0.027$ | $0.059 \pm 0.024$ |
| Ours (certain) | $0.818 \pm 0.004$ | $\mathbf{0.009} \pm 0.008$ | $\mathbf{0.028} \pm 0.020$ | $\mathbf{0.027} \pm 0.017$ |

Table 7: Comparison with different baselines on the Adult dataset using an MLP with a hidden layer of 34 units as base classifier.

- **Ground truth sensitive attribute**. We considered the case where the sensitive attribute is fully available and trained the model with fairness constraints w.r.t the ground truth. This represents the ideal situation where all the assumptions about the availability of demographic information are satisfied. This baseline is expected to achieve the best trade-offs.

- **Proxy-KNN**. Here the missing sensitive attributes are handled by data imputation using the k-nearest neighbors (KNN) of samples with missing sensitive attributes.

- **Proxy-DNN**. For this baseline, an MLP is trained on $\mathcal{D}_2$ to predict the sensitive attributes without uncertainty awareness. The network architecture used and the hyperparameter is the same as for the student in our model.

For fairness-enhancing mechanisms, we considered the Fairlean (Bird et al., 2020) implementation of the exponentiated gradient (Agarwal et al., 2018) and adversarial debiasing (Zhang et al., 2018) (Section 3). For the exponentiated gradient, we used various base classifiers including logistic regression, random forest, and gradient-boosted trees. Random forest was initialized with a maximum depth of 5 and minimum samples leaf 10, and default parameters were used for logistic regression without hyperparameter tuning. The same models and hyperparameters were used across all the datasets. Adversarial debiasing works for demographic parity and equalized odds. The architecture of the classifier, the adversary, as well as other hyperparameters used is the same as recommended by the original paper (Zhang et al., 2018). We evaluate the fairness-accuracy trade-off of every baseline by analyzing the accuracy achieved in different fairness regimes, i.e., by varying the parameter

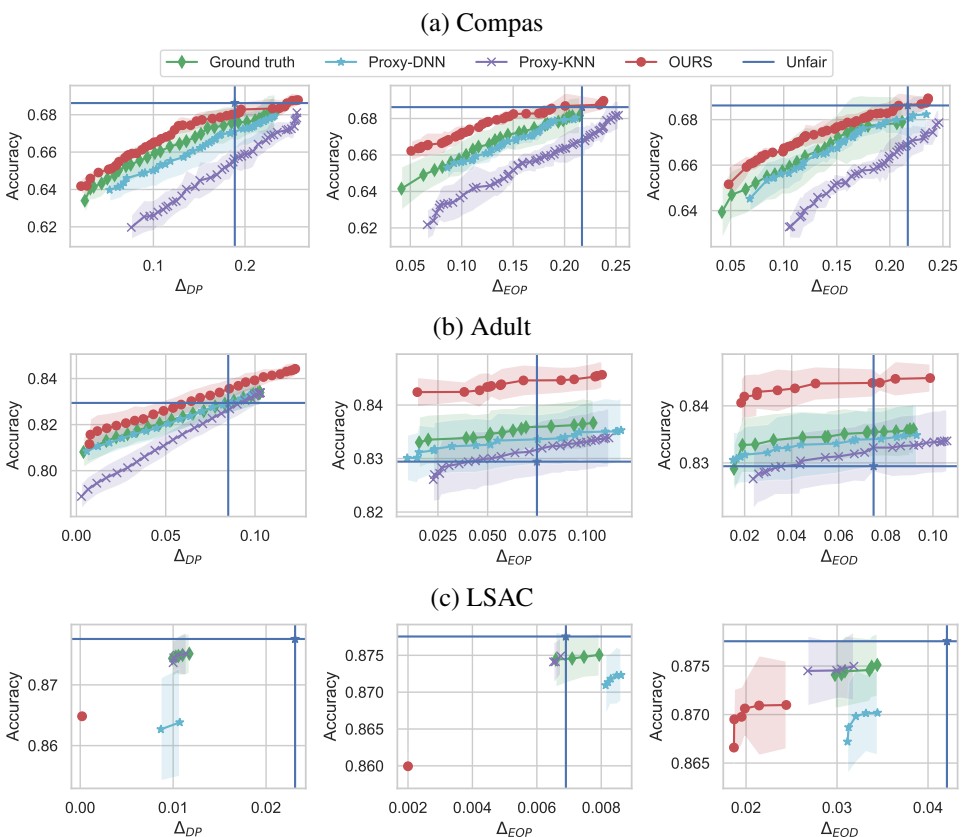

Figure 4: Accuracy-fairness trade-offs for various fairness metrics ($\Delta_{DP}$, $\Delta_{EOP}$, $\Delta_{EOD}$) and proxy sensitive attributes. Top-left is the best (highest accuracy with the lowest unfairness). Curves are created by sweeping a range of fairness coefficients $\lambda$, taking the median of 7 runs per $\lambda$, and computing the Pareto front. The fairness mechanism used is the exponentiated gradient with Random Forests as the base classifier. Shaded in the figure are the standard deviations.

$\epsilon \in [0, 1]$ controlling the balance between fairness and accuracy. For a value of $\epsilon$ close to 0, the label classifier is enforced to achieve higher accuracy while for a value close to 1 it is encouraged to achieve lower unfairness. For each value of $\epsilon$, we trained each baseline 7 times on a random subset of $\mathcal{D}_1$ (70%) using their predicted sensitive attributes, and the accuracy and fairness are measured on the remaining subset (30%), where we assumed that the joint distribution $(X, Y, A)$ is available for fairness evaluation. The results are averaged and the Pareto front is computed.

Figure 4 shows the Pareto front of the exponentiated gradient method on the Adult, Compas, and LSAC datasets using Random Forests as the base classifier. The figure shows the trade-off between fairness and accuracy for the different methods of inferring the missing sensitive attributes. From the results, we observe on all datasets and across all fairness metrics that data imputation can be an effective strategy for handling missing sensitive attributes, i.e., this fairness mechanism can efficiently improve the fairness of the model with respect to the true sensitive attributes although fairness constraints were enforced on proxy-sensitive attributes. However, we observe a difference in the fairness-accuracy trade-off for each attribute classifier. Overall, the KNN-based attribute classifier has the worst fairness-accuracy trade-off on all datasets and fairness metrics. This shows that assigning sensitive attributes based on the nearest neighbors does not produce sensitive attributes useful for achieving a trade-off close to the ground truth. While the DNN-based attribute classifier produces a better trade-off but is still suboptimal compared to the ground truth sensitive attributes. We observed similar results with different baseline models such as logistic regression and gradient-booted trees and for adversarial debiasing as the fairness mechanism. In contrast, we see that our method consistently achieves a better trade-off on all datasets and across all the fairness metrics considered. Similar results are obtained on the exponentiated gradient with logistic regression and gradient-boosted trees as base classifiers and with adversarial debiasing (see Section 6). The choice of the uncertainty threshold depends on the level of bias in the dataset, i.e. the level of information about the sensitive attribute encoded in the feature space.

Figure 6 depicts the Pareto front of various baselines on the CelebA dataset. It shows that models trained with imputed sensitive attributes via KNN consistently achieve comparable tradeoffs to models trained with fairness constraints based on the true sensitive attribute. This could be explained by the fact that gender clusters are perfectly defined in the latent space. We observed that KNN-based imputation achieved 95% accuracy the assigning the right gender. Conversely, the figures illustrate that our method outperforms baselines using both ground truth-sensitive attributes and imputation, yielding a greater number of Pareto-dominant points. This highlights the advantages of applying fairness constraints to samples with reliable sensitive attributes. Furthermore, Figure 8 (c) shows decreasing the uncertainty threshold further improves fairness while preserving the accuracy. We note the CelebA dataset can raise ethical concerns and is used only for evaluation purposes. For instance, the task of predicting the attractiveness of a photo using other facial attributes as sensitive attributes can still harm individuals even if the model predicting attractiveness is not *biased*.

## F.1 Exponentiated gradient with different baseline classifiers

Figure 5 and 7 show fairness-accuracy trade-offs achieved by the exponentiated gradient with logistic regression, and gradient-boosted trees, respectively. Similar to the results presented in the main paper, our method achieves better fairness-accuracy trade-offs.

Figure 8 shows the accuracy-fairness trade-off Exponentiated gradient using gradient-boosted trees as the base classifier for various uncertainty thresholds, the true sensitive attributes, and the predicted sensitive attributes with DNN. The results obtained are similar to random forests as the base classifier. The smaller uncertainty threshold produced the best trade-off in a high-bias regime such as the Adult dataset. While on datasets that do not encode much information about the sensitive attributes (most samples have high uncertainty) such as the New Adult and Compas datasets, the accuracy decreases as the uncertainty threshold reduces while fairness is improved or maintained. On the LSAC dataset (Figure 8(d)), we observe that increasing the uncertainty threshold results in a much higher drop in accuracy. This is explained by the fact that the average uncertainty is very high (0.66) and using a smaller threshold prunes out most of the data.

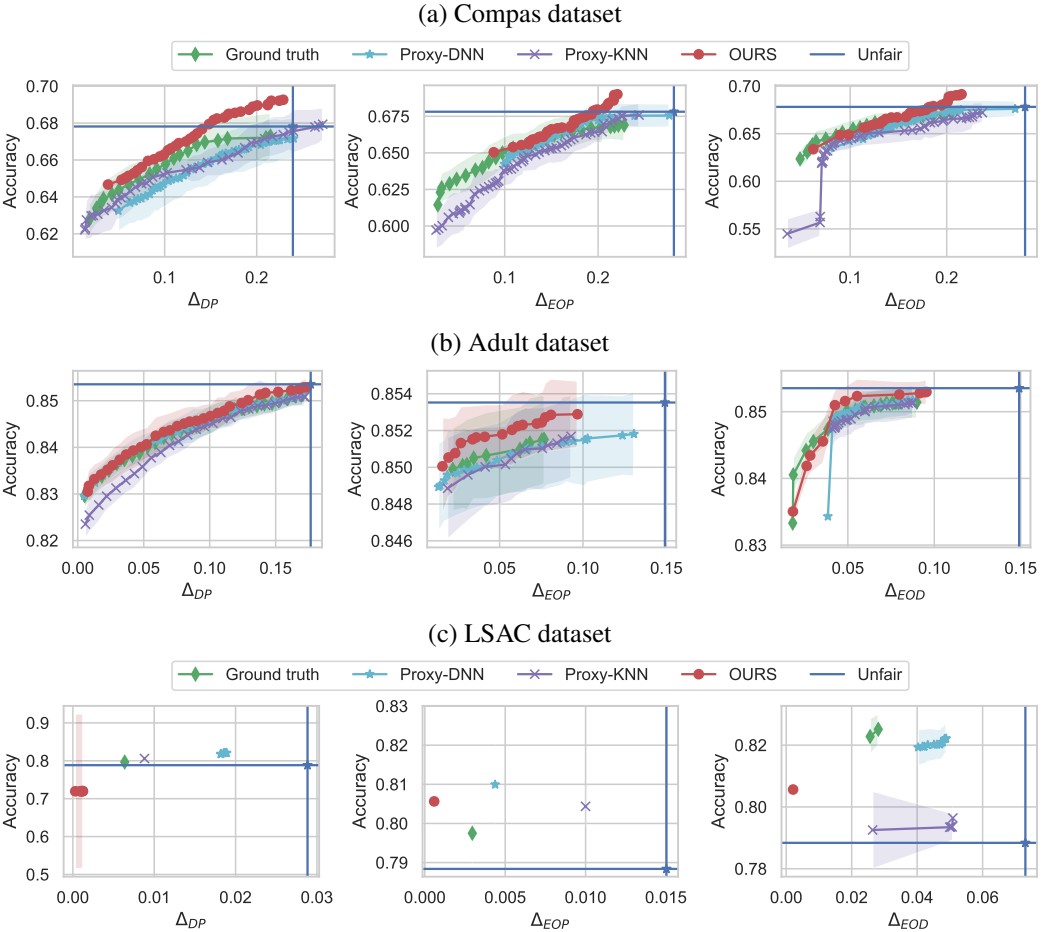

Figure 5: Accuracy-fairness trade-offs for various fairness metrics ($\Delta_{DP}$, $\Delta_{EOP}$, $\Delta_{EOD}$) and proxy-sensitive attributes. Top-left is the best (Highest accuracy with the lowest unfairness). The fairness mechanism is the Exponentiated gradient with logistic regression as the base classifier on the Compas (a), Adult (b), and LSAC (c) datasets. Shaded in the figure is the standard deviation.

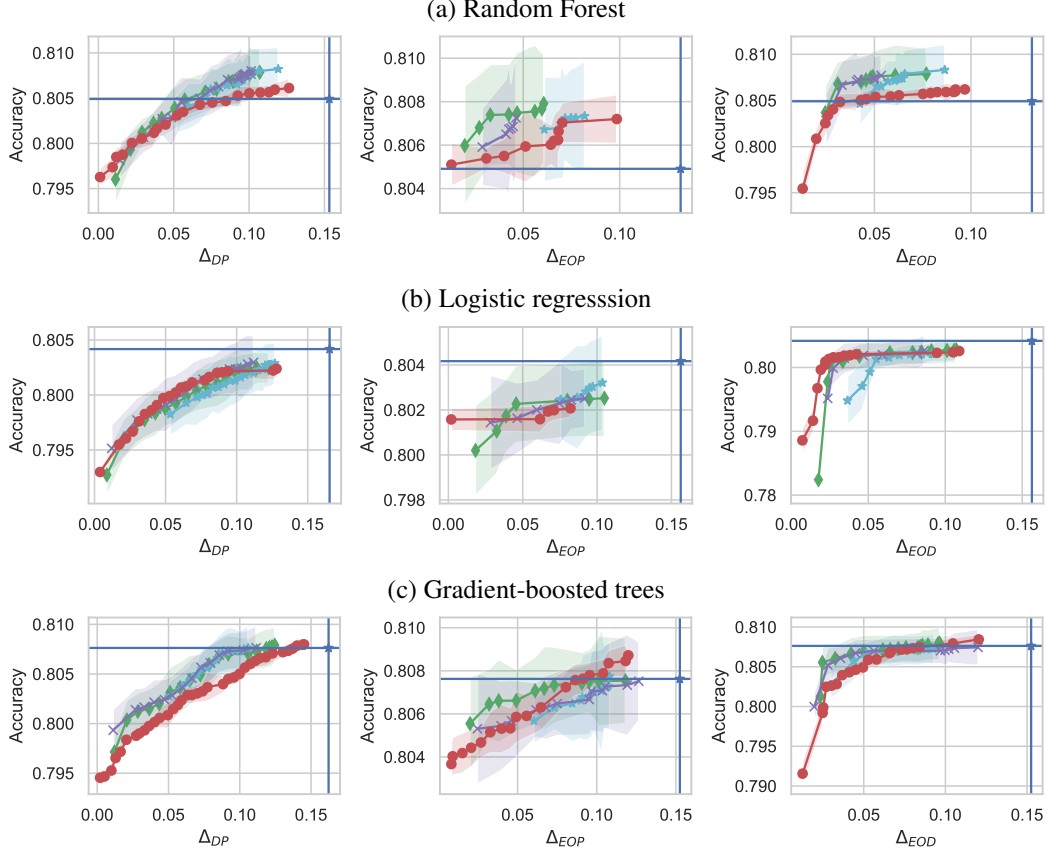

Figure 6: Exponentiated gradient with different base classifiers on the CelebA dataset: (a) random Forest, (b) logistic regression, and (c) gradient-boosted trees.

## F.2 EXPERIMENTS WITH ADVERSARIAL DEBIASING

Figure 10 shows the trade-offs for adversarial debiasing. Our methods achieve a better trade-off on the Adult datasets while for the Compas dataset, the ground-truth sensitive achieves a better trade-off. It is worth noting that adversarial debiasing is unstable to train.

## G UNCERTAINTY ESTIMATION OF DIFFERENT DEMOGRAPHIC GROUPS

In this paper, we showed that when the dataset does not encode enough information about the sensitive attributes, the attribute classifier suffers on average from greater uncertainty in the predictions of sensitive attributes. This encourages a choice of a higher uncertainty threshold to keep enough samples in order to maintain the accuracy, i.e. to prune out only the most uncertain samples. Figure 11 shows that the gap between demographic groups can increase as a smaller uncertainty threshold is used. This is explained by the fact that the model is more confident about samples from well-represented groups than samples from under-represented groups. While this gap between demographic groups can increase, our results show there are still enough samples from the disadvantaged group with reliable sensitive attributes. Thus, tuning the uncertainty threshold can result in a model that achieves a better trade-off between accuracy and various fairness metrics, by enforcing fairness constraints on samples with highly reliable sensitive attributes. Note that for the LSAC dataset, we observed the same trend. The average uncertainty is 0.66 and the minimum uncertainty is 0.62. We also observed that group representation remains consistent (35% difference) when using the average uncertainty value.

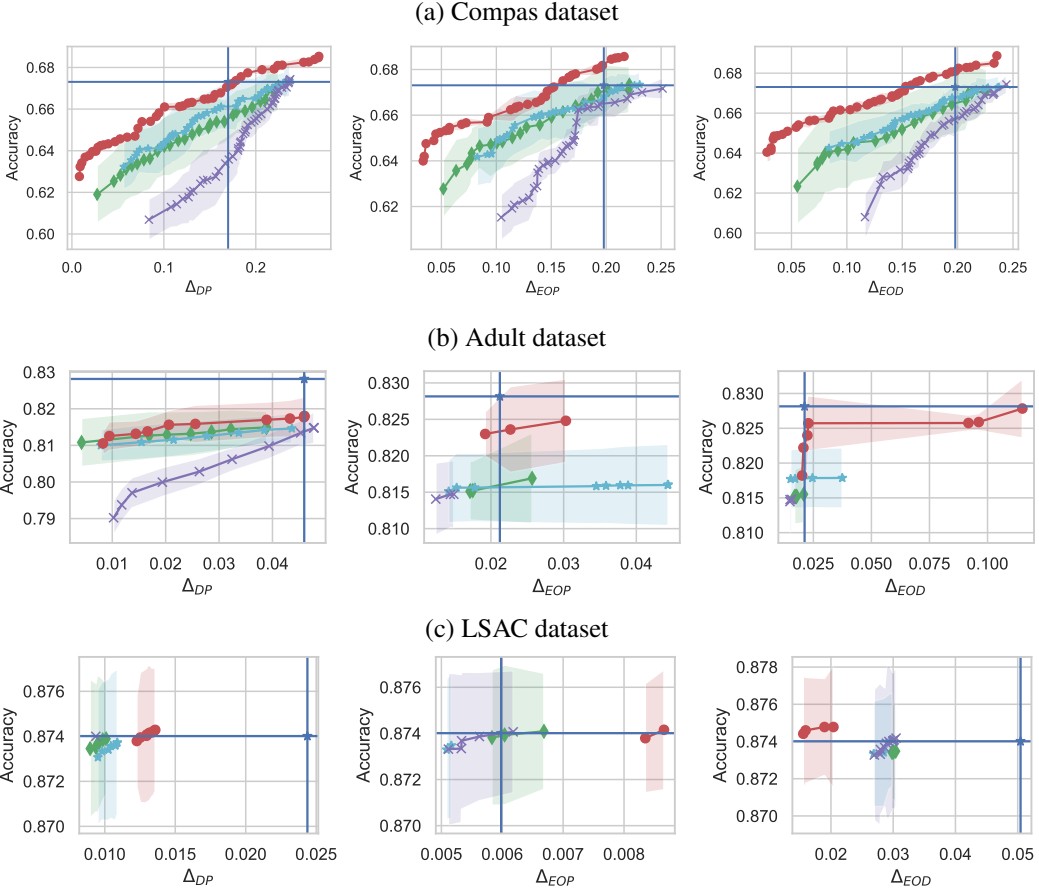

Figure 7: Accuracy-fairness tradeoffs for various fairness metrics ($\Delta_{DP}$, $\Delta_{EOP}$, $\Delta_{EOD}$) and proxy sensitive attributes. The fairness mechanism used is the Exponentiated gradient with gradient-boosted trees as the base classifier on the Compas (a) and the Adult (b) datasets. Shaded in the figure is the standard deviation.

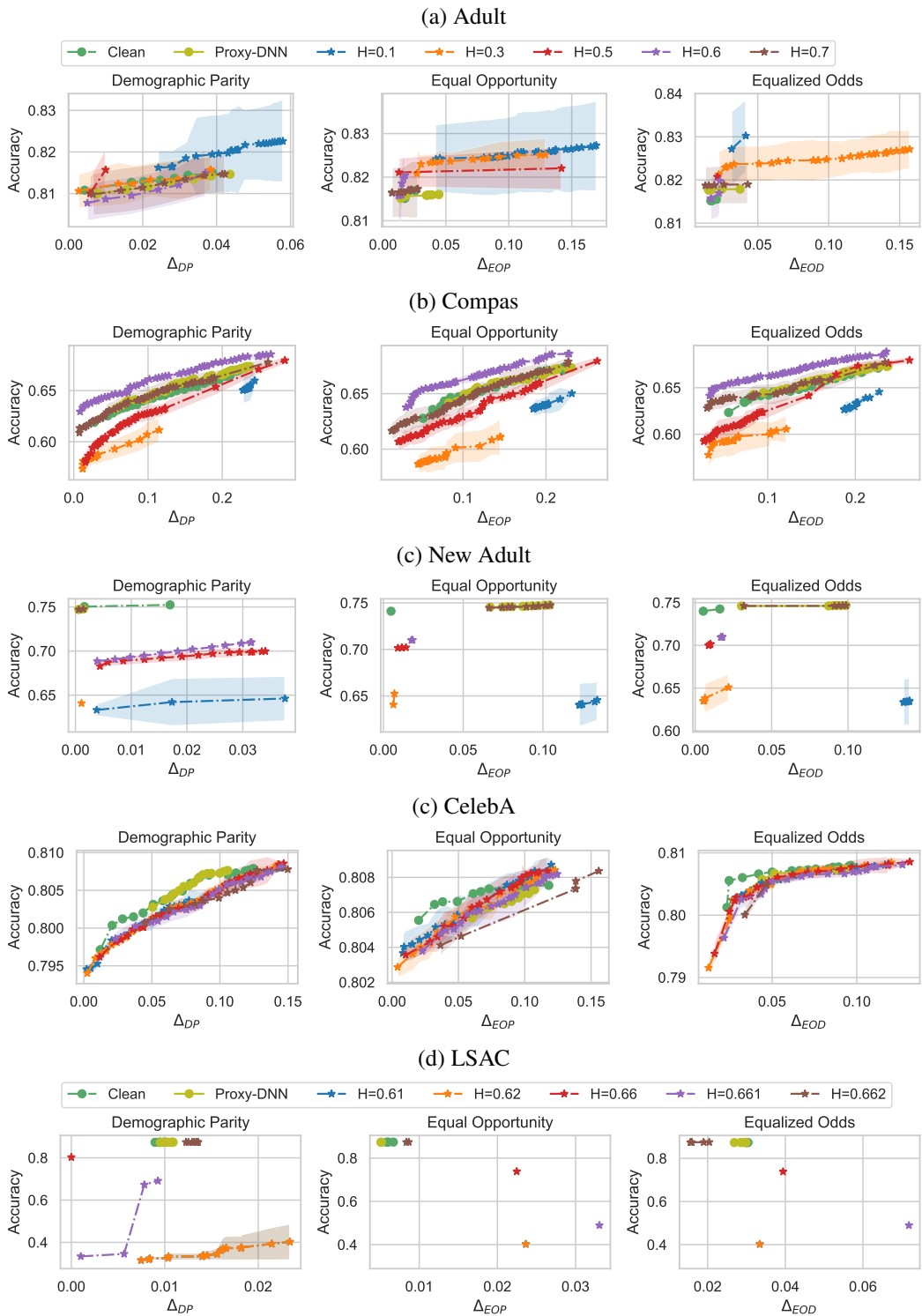

Figure 8: Exponentiated gradient with gradient-boosted trees as the base classifier. The impact of the uncertainty threshold $H$ on the fairness-accuracy trade-off for the (a) Adult, (b) Compas, (c) New Adult, (c) CelebA dataset, and (d) LSAC datasets.

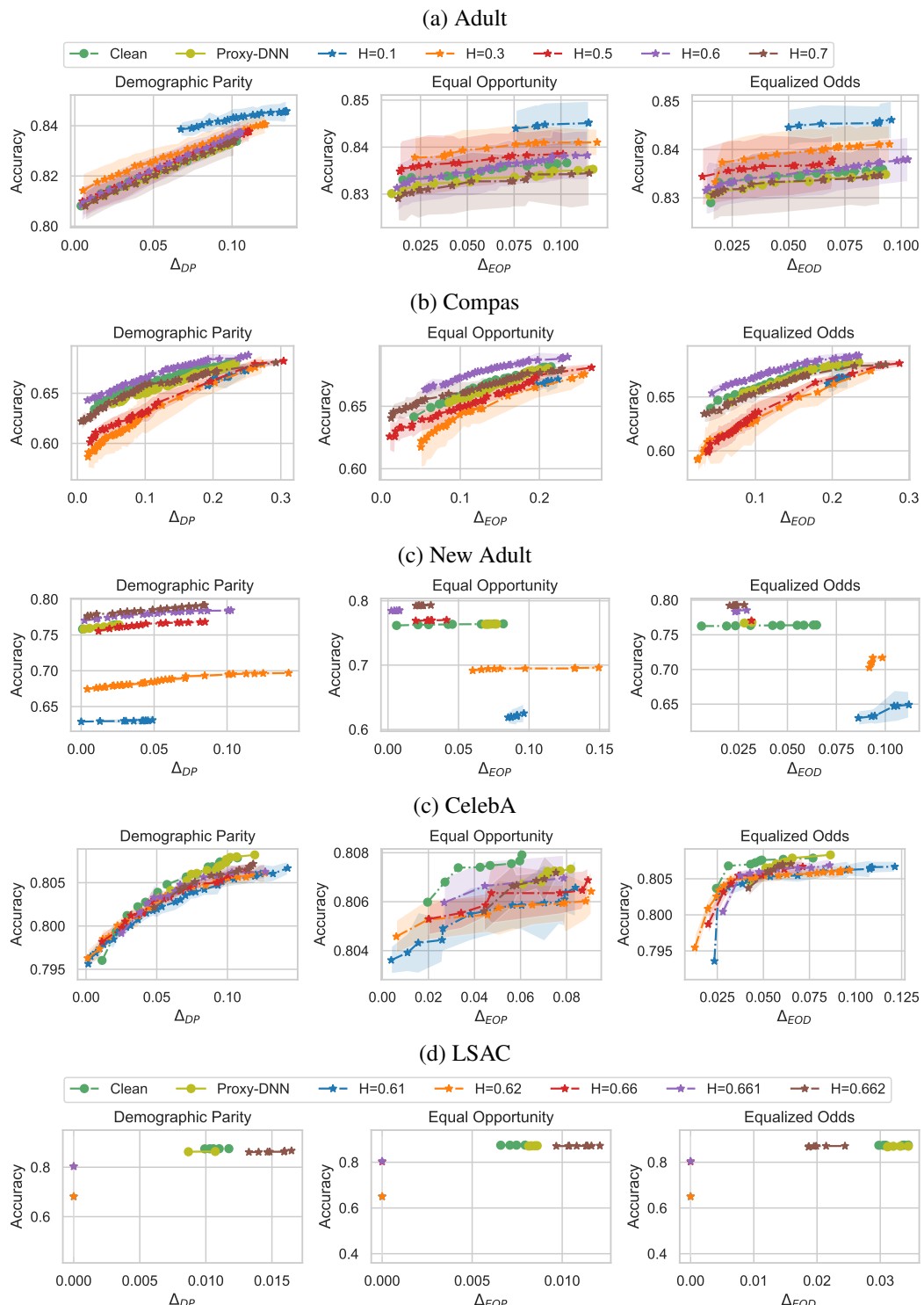

Figure 9: The impact of the uncertainty threshold $H$ on the fairness-accuracy trade-off. For the exponentiated gradient with Random Forest as the base classifier for the (a) Adult, (b) Compas, (c) New Adult, (c) CelebA dataset, and (d) LSAC datasets.

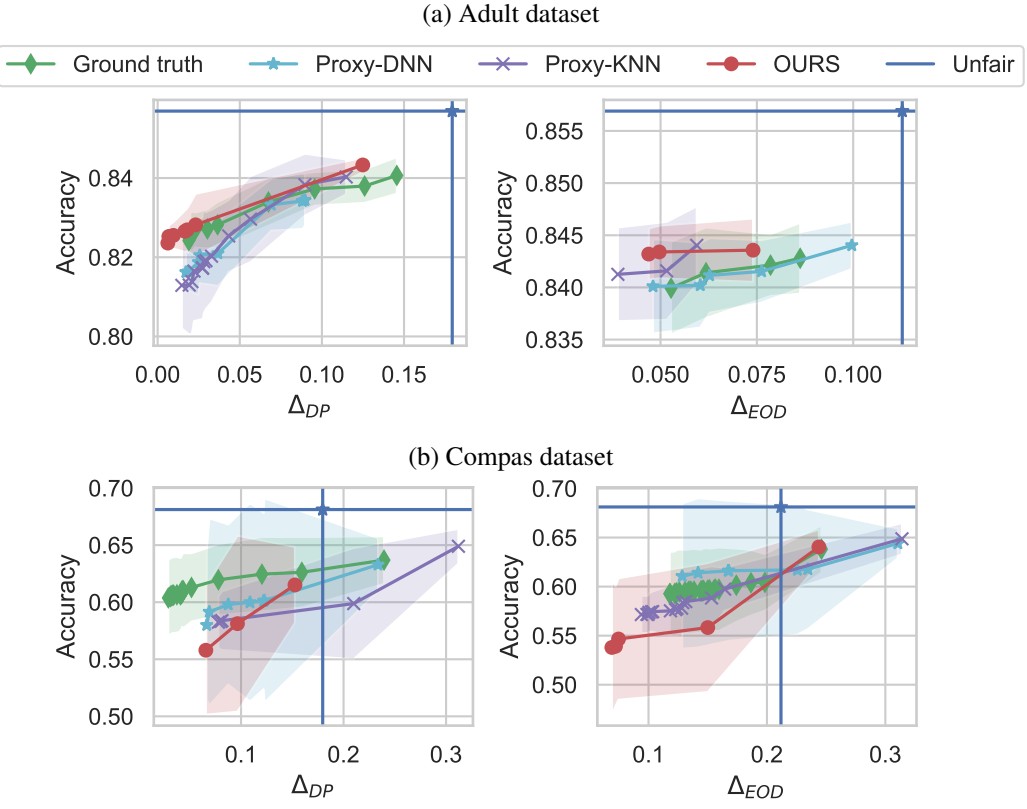

Figure 10: Adversarial debiasing. Accuracy-fairness trade-offs for various fairness metrics ($\Delta_{DP}$, $\Delta_{EOP}$) and proxy-sensitive attributes.

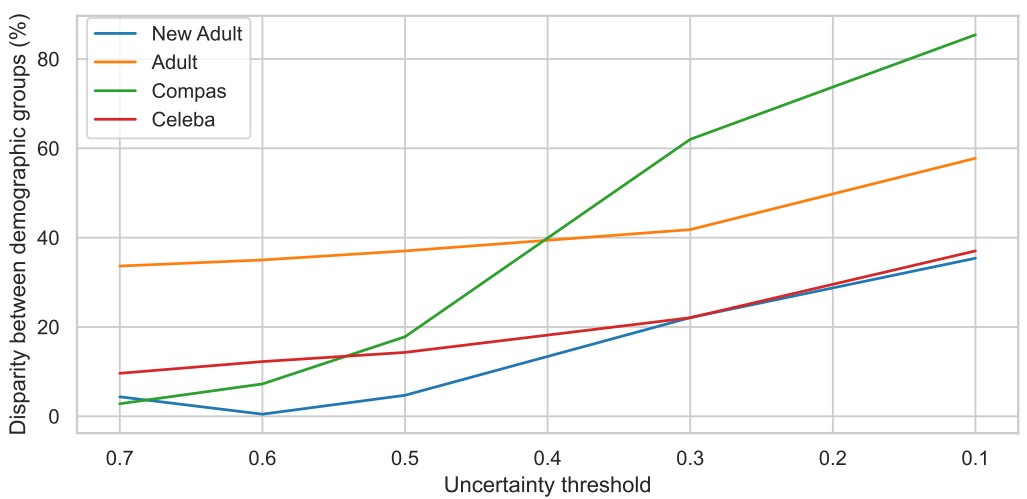

Figure 11: Demographic group representation in each dataset for different uncertainty thresholds. The gap between groups increases as the threshold becomes smaller. The plot reveals there are samples from the minority group that exhibit lower uncertainty in the prediction of their sensitive attributes.

