# OpenReview forum: "Fairness Under Demographic Scarce Regime"
_ICLR.cc/2024/Conference — ICLR 2024 Conference Withdrawn Submission_

### Official Review · Reviewer_CnjD · 2023-10-16

**Soundness:** 3 good
**Presentation:** 3 good
**Contribution:** 3 good
**Rating:** 6
**Confidence:** 4

**Summary:**

This paper presents a new solution for improving fairness under demographic scarce regime. It utilizes a teacher-student model to train an attribute classifier with uncertainty estimation first. Then trains a label classifier with only the training data whose sensitive attributes can be estimated by the attribute classifier with high confidence. Results show that the proposed approach achieves better than state-of-the-art accuracy-fairness trade-offs with three different fairness metrics on five real data sets.

**Strengths:**

1. The proposed technique is sound.

2. The presentation is clear.

3. The experiments are conducted on five real data sets.

4. The source code is provided.

**Weaknesses:**

1. The biggest concern I have is that the uncertainty threshold greatly impacts the performance of the proposed approach. It is not clear that if the authors have a reliable way to determine this threshold in practice (other than selecting the best performing one on the test data). Does tuning on a validation set reveal the best threshold for the data? Does it generalize to the test set? This should be either clarified or added as new experiments.

2. Despite the fact that the scenario discussed in this paper is not entirely new, I would still suggest the authors provide a real example scenario when D1 has no demographic information but has label information and D2 has demographic information but no label information.

**Questions:**

Q1: Results of "Ours" in Figure 2 were tuned for uncertainty threshold over [0.1, 0.7]. When you tune the threshold, are you using a validation set from the 0.7 training data or are you just selecting the best performing threshold on the 0.3 test data?

Q2: In Section 3 Problem formulation, you mentioned: " However, to be able to estimate bias in label classifier f, we assume there exists a small set of samples drawn from the joint distribution X × Y × A, i.e., samples that jointly have label and demographic information." How was this reflected in the experiments? Does it refer to the 0.3 test data you used in the experiments? If so, I do not think this is a hard requirement in real world applications.

Q3: Please clarify: in Section 4.2, "the label classifier with fairness constraints is trained on a subset D′1 ⊂ D1". Does this mean the second phase classifier does train on any information except for D'1? Is that possible to train that label classifier on the entire training set D1 with fairness constraints on D'1--- fairness loss will be 0 for data do not belonging to D'1? This could potentially increase the accuracy of the label classifier when uncertainty threshold is low.

---

> ### Author Response · Authors · 2023-11-16
> **Response to Reviewer CnjD**
>
> We thank the reviewer for detailed comments and constructive feedback.
>
> >  Despite the fact that the scenario discussed in this paper is not entirely new, I would still suggest the authors provide a real example scenario when D1 has no demographic [...]
>
> We have added in the problem formulation references to real-world examples where this setup applies. In particular, they can be found in healthcare [1, 2] or finance [3, 4] where patients/users self-report their sensitive information.
>
> > Q1: Results of "Ours" in Figure 2 were tuned for uncertainty threshold over [0.1, 0.7]. When you tune the threshold, are you using a validation set from the 0.7 training data or are you just selecting the best performing threshold on the 0.3 test data?
>
> Thank you for bringing this up. The uncertainty threshold is a hyperparameter that can be tuned on the validation set. We used 10% of the training data for validation. We highlighted in the paper how we tuned the threshold. Furthermore, we observe the higher variation of the uncertainty threshold occurs in datasets with already higher uncertainty (e.g., LSAC and New Adult), from which models without fairness constraints provide lower bias. There are similar variations in the Pareto fronts over the validation set and the best-performing finetuned threshold also transfers to the test set.
>
> >  Q2: In Section 3 Problem formulation, you mentioned: " However, to be able to estimate bias in label classifier f, we assume there exists a small set of samples drawn from the joint distribution X × Y × A, i.e., samples that jointly have label and demographic information." How was this reflected in the experiments? Does it refer to the 0.3 test data you used in the experiments? If so, I do not think this is a hard requirement in real world applications.
>
> The joint distribution X × Y × A refers to test data used in the experiments. This assumption allows us to report the true fairness violation. Moreover, we could alleviate this assumption by using proxy-sensitive attributes for the evaluation, which might be a more realistic setup. But these proxies are likely noisy and there is a risk of overestimating or underestimating the true disparities in the model. We have added a limitation section where we discuss the limitation of such assumption in real-world applications and point out the possibility of using of bias estimation approaches that effectively capture the true disparities when the proxy demographic information is used [5] [6].
>
> > Q3: Please clarify: in Section 4.2, "the label classifier with fairness constraints is trained on a subset D′1 ⊂ D1". Does this mean the second phase classifier does train on any information except for D'1? Is that possible to train that label classifier on the entire training set D1 with fairness constraints on D'1--- fairness loss will be 0 for data do not belonging to D'1? This could potentially increase the accuracy of the label classifier when uncertainty threshold is low.
>
> Great suggestion. The main objective of the paper is to provide evidence for the hypothesis we formulated. Modifying fairness-enhancing algorithms to account for the uncertainty is an interesting direction to explore. We have mentioned in the conclusion a similar perspective for future work where the fairness-enhancing model is trained on samples weighted by the uncertainty of predicting their sensitive attributes.
> Please note that all the changes in the revised version of the paper have been highlighted in blue.
>
> We hope we have addressed most of your concerns. Please consider increasing the score if you find our responses satisfactory. We would be happy to answer any further questions.
>
> [1] Brown, D. P., Knapp, C., Baker, K., & Kaufmann, M. (2016). Using Bayesian imputation to assess racial and ethnic disparities in pediatric performance measures. Health services research, 51(3), 1095-1108.
>
> [2] Fremont, A. M., Bierman, A., Wickstrom, S. L., Bird, C. E., Shah, M., Escarce, J. J., ... & Rector, T. (2005). Use of geocoding in managed care settings to identify quality disparities. Health Affairs, 24(2), 516-526
>
> [3] Zhang, Y. (2018). Assessing fair lending risks using race/ethnicity proxies. Management Science, 64(1), 178-197.
>
> [4] Silva, G. C., Trivedi, A. N., & Gutman, R. (2019). Developing and evaluating methods to impute race/ethnicity in an incomplete dataset. Health Services and Outcomes Research Methodology, 19(2-3), 175-195.
>
> [5] Chen, J., Kallus, N., Mao, X., Svacha, G., & Udell, M. (2019, January). Fairness under unawareness: Assessing disparity when protected class is unobserved. In Proceedings of the conference on fairness, accountability, and transparency
>
> [6]   Awasthi, P., Beutel, A., Kleindessner, M., Morgenstern, J., & Wang, X. (2021, March). Evaluating fairness of machine learning models under uncertain and incomplete information. In Proceedings of the 2021 ACM Conference on Fairness, Accountability, and Transparency.

---

> > ### Author Response · Authors · 2023-11-23
> > **Official Comment by Authors**
> >
> > Dear Reviewer CnjD
> >
> > We thank you for your time and constructive comments on our work which helped to improve the manuscript. We hope you could review the rebuttal where we did our best to address your concerns. In particular, in the revised version of the manuscript, we provided details on how the uncertainty threshold is tuned. We will be happy to address any remaining concerns within the remaining short time for discussion. We kindly ask you to adjust the score if you found our responses satisfactory.

---

### Official Review · Reviewer_EiNj · 2023-10-30

**Soundness:** 1 poor
**Presentation:** 3 good
**Contribution:** 2 fair
**Rating:** 3
**Confidence:** 4

**Summary:**

The authors introduce a framework designed to facilitate the training of fairness-enhancing interventions when sensitive information is only partially observed. Their approach involves developing a classifier that seeks to predict the sensitive attributes of instances. Subsequently, they leverage instances with the least uncertain predictions, along with their predicted sensitive attributes, to train the fairness-enhancing intervention.

**Strengths:**

S1 - The authors present a solution to a significant challenge that fairness-enhancing interventions may encounter when implemented in real-world applications.

S2 - The experiments exhibit several strengths, including the diverse range of classification tasks involving different datasets, the validation of various aspects of the work. Particularly noteworthy are the investigations into the relationship between the threshold and the encoding of sensitive information by features, as well as the analysis of uncertainty in the sensitive attribute and the impact on the fairness of non-fairness-aware classifiers. Overall, the experiments provide supporting evidence for their central hypothesis.

S3 - The paper's commitment to reproducibility is highly commendable. The detailed and transparent presentation of the experimental setup, data sources, and code availability significantly enhances the reliability and trustworthiness of the research findings. This transparency not only promotes the understanding of the study but also encourages further research in the field.

S4 - The paper's writing is remarkably clear, making it easy for readers to grasp the content. Furthermore, the well-structured sections and the logical flow of information make it easy for readers to follow the research from start to finish.

S5 - The paper effectively incorporates citations of pertinent related works, which helps contextualize their approach within the existing literature.

**Weaknesses:**

W1 - I believe there's a significant ethical concern in constructing a classifier with the objective of predicting the sensitive attribute of instances. This practice may raise legal and ethical issues, especially when individuals choose not to disclose this information willingly. Instead, it would be preferable if this classifier incorporated desirable privacy properties, as outlined in Diana et al. (2022).

W2 - I find the comparison with respect to the state of the art to be lacking. The attribute classifiers chosen in this work, such as Proxy-kNN and Proxy-DNN, are rather simplistic and not well-documented in existing literature. Moreover, there are established attribute classifiers like those introduced by Diana et al. (2022) and Awasthi et al. (2021) that are not considered in this comparison.
Furthermore, the selected methods for the 'baselines' (Lahoti et al., 2020; Hashimoto et al., 2018; Levy et al., 2020; Yan et al., 2020; Zhao et al., 2022) assume that they lack access to the sensitive attribute, making the experimental setting fundamentally different. Therefore, comparing the proposed approach with these state-of-the-art methods that operate under distinct conditions may not provide a fair assessment of its improvements and contributions to the field.
To offer a more comprehensive evaluation of the proposed method and better understand its advancements over existing techniques, I suggest including experiments involving Diana et al. (2022) and Awasthi et al. (2021).

W3 - The authors assert in the abstract that 'our framework outperforms models trained with constraints on the true sensitive attribute,' referring to the results from Figure 2. However, this result only considers a single fairness-enhancing intervention. Additionally, their framework does not consistently outperform models trained with ground truth sensitive attribute values in all cases, making the statement partially true. This discrepancy is even more apparent in the results from Figure 9, where a different fairness-enhancing intervention is employed. It's unclear to what extent this outcome is influenced by the chosen fairness-enhancing intervention. Both interventions analyzed in the study share similarities, and it would be beneficial to examine a more diverse set of fairness-enhancing interventions to better understand the impact of the chosen intervention on the results. Therefore, I recommend that the authors modify the statement to emphasize that the framework outperforms models trained with constraints on the true sensitive attribute in some cases, and I encourage them to delve into the conditions under which this outperformance occurs.

W4 - I believe that the related works section should also encompass privacy-related research. There are privacy-focused approaches, such as cryptographic solutions, that deal with situations where sensitive features are available but can only be accessed through secure cryptographic methods. Veale and Binns [1] or Kilbertus et al. [2] discuss scenarios where individuals' sensitive information is held by a third party or the individuals themselves, respectively, and can only be accessed via secure multiparty computation. Additionally, Jagielski et al. [3] explores cases in which sensitive features can only be utilized in a differentially private manner. Considering the nature of inferring sensitive information, privacy considerations become crucial. Therefore, it would be valuable to include these privacy-focused works in the related literature to provide a more comprehensive perspective.

[1] Veale, M., & Binns, R. (2017). Fairer machine learning in the real world: Mitigating discrimination without collecting sensitive data. Big Data & Society, 4(2), 2053951717743530.

[2] Kilbertus, N., Gascón, A., Kusner, M., Veale, M., Gummadi, K., & Weller, A. (2018, July). Blind justice: Fairness with encrypted sensitive attributes. In International Conference on Machine Learning (pp. 2630-2639). PMLR.

[3] Jagielski, M., Kearns, M., Mao, J., Oprea, A., Roth, A., Sharifi-Malvajerdi, S., & Ullman, J. (2019, May). Differentially private fair learning. In International Conference on Machine Learning (pp. 3000-3008). PMLR.

**Questions:**

Q1 - The experiments provide support for the assertion that discriminating against samples with more uncertain sensitive information is a challenging task. Rather than attempting to predict the sensitive information of instances (an action that is illegal and morally questionable) and use those instances for which you know the sensitive information with high confidence, why not directly utilize those instances for which the uncertainty is highest with respect to the sensitive attribute and train a non fairness-aware classifier on top of those instances? In other words, perhaps utilizing your attribute classifier to identify the most 'fair' samples based on high uncertainty in sensitive information might yield more ethically favourable results.

Q2 - For classifiers that propose fairness-enhancing interventions while lacking information on the specific sensitive attribute considered in the experimental section (Lahoti et al., 2020; Hashimoto et al., 2018; Levy et al., 2020; Yan et al., 2020; Zhao et al., 2022), it's essential to clarify the dataset utilized. Do you feed them with the complete D1, D1 + D2, or only D1'?

Q3 - The experiments demonstrate significant variations in results depending on the chosen uncertainty threshold. If this model were to be applied in a real-world scenario, do you have a practical method for selecting the optimal uncertainty threshold, rather than relying on trial and error to determine the best-performing value?

---

> ### Author Response · Authors · 2023-11-16
> **Response to reviewer EiNj - Part 1**
>
> We thank the reviewer for their comments and feedback. Below we provide clarifications for the concerns raised.
>
> > W1 I believe there's a significant ethical concern in constructing a classifier with the objective of predicting […] Instead, it would be preferable if this classifier incorporated desirable privacy properties, as outlined in Diana et al. (2022)
>
> We acknowledge the ethical concern in inferring sensitive attributes. This represents a general concern for the group of methods relying on proxy-sensitive information. We believe the prediction of sensitive attributes with the objective of mitigating discrimination in high-stakes decision-making scenarios could be ethically acceptable. The inference of the sensitive attributes—using our proposed method or others, should not be used for a purpose different from bias assessment and mitigation.  Our method provides insights into the benefit of uncertainty-awareness in predicting sensitive attributes for fairness of the downstream classifiers. Furthermore, the ethical concern in inferring sensitive attributes comes from the dilemma posed between privacy and fairness. In particular, laws or regulations enforce discrimination-free decision-making while they also prohibit the use or collection of demographic information, which is necessary for auditing and mitigating discrimination. While privacy guarantees for the sensitive attributes are out of the scope of the paper incorporating privacy aspects is a suggestion that we welcome. Moreover, methods designed under a privacy-preservation setup generally do not guarantee that an adversary cannot reconstruct the sensitive attributes, especially for methods relying on trusted third parties or secure multi-party computation.  For example,  [1] shows that one can exploit information about a fair model to reconstruct the sensitive attributes, even with black box access.  We have added a limitation section in the revised version (please see Appendix A) where we discuss ethical concerns in inferring sensitive information.
>
> > W2 I find the comparison with respect to the state of the art to be lacking. The attribute classifiers chosen in this work, such as Proxy-kNN and Proxy-DNN, are rather simplistic and not well-documented in existing literature [...]
>
> Some of the baselines considered also use proxy-sensitive attributes (Zhao et al., 2022; Liang et al., 2023; Yan et al., 2020;), and methods without relying on sensitive attributes such as ARL (Lahoti et al., 2020) can also improve group metrics such as equalized odds. Other baselines are considered for a comprehensive evaluation and we highlighted that the experimental setup might be different, in particular for methods targeting worst-case group improvement.
> We couldn’t perform comparisons with Diana et al, (2022) as the authors did not publish their code and we couldn’t reproduce their results within the short rebuttal period. Moreover, Awasthi et al. (2021) focused on postprocessing techniques for equalized odds, which generally highly impact accuracy,  while we considered inprocessing techniques with controllable fairness-accuracy tradeoffs. Furthermore, the results reported using true sensitive attributes represent the optimal expected baseline for fairness, and comparison shows our method can achieve similar and in most cases better tradeoffs. It does not seem to be the case for Awasthi et al. (2021) and Diana et al, (2022) on the same datasets.
>
> > W3 The authors assert in the abstract that 'our framework outperforms models trained with constraints on the true sensitive attribute,' referring to the results from Figure 2. However, this result only considers a single [...]
>
> Thank you for the suggestions. We have updated the statement accordingly and included privacy-related approaches in the related work section in the revised version.  We have also highlighted in the limitation section and emphasized the need for further studies to explore whether our method and hypothesis extend to other fairness-enhancing methods, e.g., preprocessing or post-processing techniques.
>
> [1] Ferry, J., Aïvodji, U., Gambs, S., Huguet, M. J., & Siala, M. (2023, February). Exploiting Fairness to Enhance Sensitive Attributes Reconstruction. In 2023 IEEE Conference on Secure and Trustworthy Machine Learning (SaTML)

---

> > ### Author Response · Authors · 2023-11-16
> > **Response to reviewer EiNj - Part 2**
> >
> > > Q1 - The experiments provide support for the assertion that discriminating against samples with more uncertain sensitive information is a challenging task [...] why not directly utilize those instances for which the uncertainty is highest with respect to the sensitive attribute and train a non fairness-aware classifier on top of those instances? [...]
> >
> > Indeed, it is possible to use samples with the highest uncertainty in the sensitive attribute predictions to train models without fairness constraints. Table 1 in the main paper shows that datasets with higher uncertainty in the sensitive attribute prediction tend to be inherently fairer.
> > Furthermore, we performed additional experiments where we trained different classifiers (Logistic Regression and Random Forest) without fairness constraints but using instances with higher uncertainty and for different uncertainty thresholds (see Appendix G).
> > Results are presented in the newly added Figure 11 of the revised version of the paper. The results show we can get fairer models by pruning out samples with low uncertainty in attribute prediction. This shows that our method can also be applied in settings where the use of inferred protected attributes is illegal or ethically concerning.
> >
> > > Q2 - For classifiers that propose fairness-enhancing interventions while lacking information on the specific sensitive attribute considered in the experimental section [...] Do you feed them with the complete D1, D1 + D2, or only D1'?
> >
> > We feed all the baselines on the complete D1, i.e., the dataset with the target labels and missing sensitive attributes. For the baselines using a proxy classifier the dataset D2 is used to train to proxy classifier, which is used to infer the missing sensitive attributes in D1. Only our method uses D1' where the uncertainty threshold was applied. Overall, we only use D1 for training each baseline, or D1 with the inferred sensitive attributes where needed. We have clarified this in the revised version of the paper.
> >
> > > Q3 - The experiments demonstrate significant variations in results depending on the chosen uncertainty threshold [...]
> >
> > The uncertainty threshold is a hyperparameter that can be finetuned on a validation set. A similar variation is also reflected in the validation set and the results show that the best-performing threshold on the validation set transfers well to the test set. Furthermore, the high variation of the uncertainty threshold occurs in datasets with already higher uncertainty (e.g., LSAC and New Adult). We referred to this setting as a _low bias regime_, i.e., a situation where most samples already have high uncertainty of the sensitive attributes and the model already has low bias.  We have highlighted how we tuned the uncertainty threshold in the experimental setup.
> >
> > All the changes in the revised version of the paper are highlighted in blue.
> >
> > We hope we could address most of your concerns. Kindly consider increasing the score if you find our responses satisfactory. We would be happy to answer any further questions.

---

> ### Comment · Reviewer_EiNj · 2023-11-17
> **Response to rebuttal**
>
> I appreciate your response addressing my concerns.
>
> Linked to W1, I acknowledge that legal regulations mandate discrimination-free decision-making while simultaneously prohibiting the use or collection of demographic information. However, I find this justification insufficient for supporting a practice that is both illegal and ethically dubious, involving the inference of sensitive information from individuals who have chosen not to disclose such details. Particularly considering the wealth of contributions in the field of differential privacy, it seems plausible to explore alternative approaches. I concur with reviewer PEk1 that the current approach appears somewhat basic, and the incorporation of privacy-preserving techniques could significantly enhance the paper's potential.
>
> Concerning W2, I believe the comparison of the proposed method against state-of-the-art methods should be prominently featured in the main text to underscore the advantages of using this approach over existing methods. This emphasis is especially crucial given the reported results indicating a significant outperformance of the proposed method compared to the considered state-of-the-art approaches. Consequently, the comparisons involving Proxy-kNN and Proxy-DNN could be relegated to the appendix.
>
> Furthermore, I'd like to take this opportunity to provide feedback on a couple of potential improvements in Tables from Appendix D and E. For instance, in Tables 3 and 4, Vanilla emerges as the most accurate method, but it is not highlighted in bold; instead, ARL and your method are highlighted. Additionally, it would be beneficial to maintain consistency in the number of decimals reported in the results. In Tables 5 and 6, most numbers have 3 decimals, but a few have 4 decimals.

---

> > ### Author Response · Authors · 2023-11-20
> > **Reply to reviewer EiNj's feedback.**
> >
> > We thank the reviewer for the feedback and the additional suggestions for improvement.
> >
> > Regarding the integration of privacy-preserving mechanisms. The definition of differential privacy (whether local or centralized) assumes the availability of personal data on which an algorithm will operate. We are concerned with a scenario where the personal data that can help enforce fairness constraints is unavailable. Therefore, using differential privacy or any other privacy-preserving technique would be irrelevant. However, we agree with the reviewer on the fact that inferring sensitive attributes could pose ethical risks in cases where such a classifier is used for other purposes. We will discuss that aspect in the paper as well as how DP can be used to prevent it.
> > Furthermore, our framework also provides an alternative. As suggested by the reviewer in the initial review, we showed that our method can be applied to train fair models without using predicted sensitive attributes. To demonstrate that, we trained non fairness-aware algorithms on top of instances with the highest uncertainty in sensitive attributes (Please see Figure 3 in the Appendix). This alternative does not use predicted sensitive information but the uncertainty of the predictions provided by our framework.  We believe this alternative approach can be more ethically acceptable.
> >
> > The reviewer PEk1 believes our method is basic because it performs transfer learning and thresholding of the predictions, which is not the case. As requested by reviewer PEk1, we clarified the misunderstanding and highlighted that the proposed approach is not about transfer learning, but fair training with uncertainty-awareness, which itself is novel.
> >
> > We have moved the comparison against state-of-the-art methods in the main text and fixed the inconsistencies in the tables. The bolded values represent the best-performing baselines among fairness-enhancing methods without (full) demographic information.
> >
> > We thank the reviewer again for the comments. Kindly consider increasing the score if we have addressed most of your concerns. We will be happy to provide clarifications for any further concerns.

---

> > > ### Comment · Reviewer_EiNj · 2023-11-21
> > > **Reply to authors answer**
> > >
> > > Thank you once again for addressing my concerns.
> > >
> > > I still have a couple of points to discuss:
> > >
> > > (a) While Figure 3 in the Appendix is informative, it would be more insightful if it included additional baseline methods mentioned in the experimental section (e.g., FairDA, DRO).
> > >
> > > (b) Based on your results, training the classifier on instances with high uncertainty in sensitive information yields excellent outcomes (Table 6). However, a more comprehensive comparison of both methods is lacking. For Adult, COMPAS, and LSAC, you apply the approach involving instances with certain sensitive information, while evaluating the CelebA classification tasks with the method that considers instances with uncertain sensitive information. It would be beneficial to assess both approaches for all four classification tasks in Tables 1, 2, 3, and 6 as OURS(certain) and OURS(uncertain). This way, if OURS(uncertain) performs comparably to OURS(certain) and surpasses all baselines, it could offer a fair method without the need to infer sensitive information, avoiding any illegal or morally questionable tasks. Additionally, exploring the conditions under which one method might outperform the other would provide valuable insights.
> > >
> > > (c) I concur with Reviewer PEk1 that it seems unusual for the proposed method to outperform all baselines by such a significant margin. Providing convincing reasons for this observed performance would be appreciated.
> > >
> > > (d) In Table 6, there is still a need to address the decimal problem in the row for 'ours' in the first column.

---

> > > > ### Author Response · Authors · 2023-11-22
> > > > **We provide more clarifications**
> > > >
> > > > We thank the reviewer for their comment and for engaging in the discussion.
> > > >
> > > > >(a) While Figure 3 in the Appendix is informative, it would be more insightful if it included additional baseline methods mentioned in the experimental section (e.g., FairDA, DRO).
> > > >
> > > > Figure 3 shows an ablation study of the impact of the uncertainty threshold over the fairness of a model trained without fairness constraints. For this, we considered different uncertainty thresholds, i.e., 0.0, 0.1, …, 0.6, (in the x-axis), and reported the fairness and accuracy for each threshold. We couldn’t include other baselines in Figure 3 as they do not depend on the uncertainty threshold.  However, in the revised version of the paper, we included the alternative approach (Ours (uncertain)) as a baseline in Tables 1, 2, 3, and 6 to compare with other baselines.
> > > >
> > > > >(b) Based on your results, training the classifier on instances with high uncertainty in sensitive information yields excellent outcomes (Table 6).
> > > >
> > > > Sorry for the confusion. Table 6 follows the same setup as tables 1, 2, and 3, where we apply our approach involving fairness constraints on instances with certain sensitive information. Table 6 was relocated to the Appendix due the to page limit.
> > > > Furthermore, we have compared both approaches, OURS(certain) and OURS(uncertain) by adding OURS (uncertain) to all tables in the revised version of the paper. For OURS (uncertain) the uncertainty threshold was tuned on the validation set, and the highest value (0.6) was the best-performing. The results show that OURS(uncertain)  can outperform other baselines in datasets where the average uncertainty of the sensitive information is low, e.g., Adult and CelebA datasets, while providing comparable results on other datasets having higher uncertainty (LSAC and Compas datasets). For example, the LSAC dataset has an average uncertainty of 0.66 which is close to the maximum uncertainty (ln2)  meaning most samples on this dataset already have uncertain sensitive information and the unfairness is already low. As no fairness constraints are enforced in OURS (uncertain), most data samples are preserved after applying the uncertainty threshold, thus having less impact on fairness and accuracy. We have highlighted the conditions under which this method (OURS (uncertain)) can outperform other baselines in the revised version.
> > > >
> > > > >(c) I concur with Reviewer PEk1 that it seems unusual for the proposed method to outperform all baselines by such a significant margin. Providing convincing reasons for this observed performance would be appreciated.
> > > >
> > > > The performance gap is justified by the performances of the underlying fairness-enhancing technique used by our approach (Exponentiated Gradient [2]). Exponentiated Gradient offers an effective way to control fairness violations. For example, with the fairness violation set to 0, the method can derive a classifier that does not exceed the fairness violation, at least with some small margins, while minimizing the classification error. In Table 6, Exponentiated Gradient is the baseline method that uses the true sensitive attributes (Vanilla (with fairness)), with fairness violation set to 0 (strict fairness) and it already provides a significant gap compared to other baselines on this dataset (DP: 0.008; EOP:  0.018, OED: 0.017). Applying our method, i.e., using Exponentiated Gradient as the fairness enhancing method but applying fairness constraints over samples with more certain sensitive attributes, improves the tradeoffs. These results follow our hypothesis and explain the gap compared to other baselines. Moreover, methods that aim for Rawlsian max-min fairness (e.g., DRO, ARL, CVarDRO) generally do not provide significant improvement in group fairness metrics. We have fixed the inconsistency in Table 6.
> > > >
> > > > We thank the reviewer again for the engaging discussion and hope we addressed all the concerns. We will be happy to provide more clarification where needed.

---

> ### Comment · Reviewer_EiNj · 2023-11-22
> **Additional question on results**
>
> I appreciate your additional clarifications regarding Figure 3 and the superiority of your results, and thank you for rectifying my misunderstanding on Table 6. Furthermore, I find it noteworthy that Ours(uncertain) yields promising results.
>
> I have an additional question: You assert that the superior performance of your results might be attributed to the utilization of a highly potent fairness-enhancing intervention (exponentiated gradient). Nevertheless, the remaining baselines employ logistic regression as their foundational classifier, a relatively simplistic approach that may not be the most suitable choice for these classification tasks. I'm inclined to think that this setup doesn't facilitate a truly fair comparison. Have you explored the use of other base classifiers such as SVM or even some straightforward neural networks?

---

> > ### Author Response · Authors · 2023-11-22
> > **Reply to the additional question**
> >
> > We thank the reviewer for their response and appreciate the constructive discussion.
> >
> > Our method also uses Logistic Regression as the base model with Exponentiated Gradient as the fairness mechanism, thus making the comparison with other baselines fair as all the methods use the same model class, i.e., Logistic Regression. Furthermore, Exponentiated Gradient is a generic algorithm that can be used with different types of models such as SVM or feed-forward networks.
> >
> > We considered more complex base models such as Random Forest and Gradient Boosted Trees. The results in the Appendix show that our proposed method can provide a significant improvement in fairness accuracy tradeoffs when using more complex and non-linear models.  In particular, Figures 8 and 9, show that OURS (certain) can yield Pareto dominant points with fairness violation closer to 0 across different datasets. Figure 3(b) shows that applying OURS (uncertain) on Random Forest can significantly improve fairness with less impact on accuracy (<5% drop across datasets). These results suggest a similar performance gap with other baselines under more complex models.
> >
> > We hope our responses addressed all the concerns of the reviewer and we will be happy to provide more clarifications.

---

> ### Author Response · Authors · 2023-11-23
> **Additional results**
>
> Dear Reviewer EiNj,
>
> Thank you for your time and constructive comments. We hope our last response provided a clarification to your question.
> As per your suggestion, we explored the use of a different base classifier such as a feedforward network. We could not consider SVM as all of the methods considered for comparison do not support SVM.
> We trained all the baselines using a multi-layer perception of one hidden layer with 32 units. We evaluated each method on the Adult dataset under the same setup to analyze its impact on the comparison. The results on other datasets are not provided due to the limited time and computational resources. The newly added Table 7 in the appendix shows that our method can still outperform other baselines under a more complex non-linear base classifier. We observed an increase in the accuracy of other baselines due to the increased capacity of the neural network, while our method still provides a significant gap in fairness.
>
> We will be happy to address any remaining concerns within the remaining short time for discussion. We would like to ask you to kindly consider adjusting your score if you find that our discussion addressed your concern.

---

### Official Review · Reviewer_PEk1 · 2023-10-31

**Soundness:** 2 fair
**Presentation:** 3 good
**Contribution:** 1 poor
**Rating:** 3
**Confidence:** 2

**Summary:**

The paper studies how to achieve a better fairness-accuracy tradeoff when no access to full sensitive attributes in the dataset. The method has two steps: (1) training a proxy classifier to predict the missing sensitive attributes with a student-teacher distillation and (2) thresholding the confidence on predictions. The paper evaluates the method on common fairness benchmark datasets.

**Strengths:**

The paper targets an important problem. Given the increasingly stringent privacy constraint, the problem of studying fairness without full access to sensitive attributes is an important problem.

**Weaknesses:**

I have two major concerns.

(1) If I am not mistaken, it seems the technical contribution of the paper is limited. The first step is not far from merely training a classifier to predict sensitive attributes, which is usually treated as a baseline in this area, with a little enhancement of student-teacher transfer learning. Overall, I do not see significant technical novelty. The second step is just to filter by thresholding prediction confidence. I have a hard time finding the technical contributions of the paper.

(2) In experiments, the paper only compares to the basic bias mitigation algorithm, but there is literature of fairness with not full access to the sensitive attributes:

[1] Diana, Emily, et al. "Multiaccurate proxies for downstream fairness." Proceedings of the 2022 ACM Conference on Fairness, Accountability, and Transparency. 2022.

[2] Chen, Jiahao, et al. "Fairness under unawareness: Assessing disparity when protected class is unobserved." Proceedings of the conference on fairness, accountability, and transparency. 2019.

[3] Prost, Flavien, et al. "Measuring model fairness under noisy covariates: A theoretical perspective." Proceedings of the 2021 AAAI/ACM Conference on AI, Ethic

[4] Fogliato, Riccardo, Alexandra Chouldechova, and Max G’Sell. "Fairness evaluation in presence of biased noisy labels." International conference on artificial intelligence and statistics. PMLR, 2020.

[5] Zhu, Zhaowei, et al. "Weak Proxies are Sufficient and Preferable for Fairness with Missing Sensitive Attributes." International Conference on Machine Learning, 2023.

[6] Yan, Shen, Hsien-te Kao, and Emilio Ferrara. "Fair class balancing: Enhancing model fairness without observing sensitive attributes." Proceedings of the 29th ACM International Conference on Information & Knowledge Management. 2020.

I do not see why this paper should not be compared with any of those works.

**Questions:**

1. Can authors clarify if there is anything I misunderstood about the technical contribution? Note that there is no point in merely repeating the details of the method. The constructive communication would be to point out if I am wrong when I say the method is just training a proxy classifier with transfer learning and thresholding predictions.

2. Can authors explain the reason why no comparison to any of the methods in the literature of fairness without full access to sensitive attributes?

---

> ### Author Response · Authors · 2023-11-15
> **Response to reviewer PEk1**
>
> We thank the reviewer for the feedback on our work. Below we address the concerns.
>
> > Can authors clarify if there is anything I misunderstood about the technical contribution? Note that there is no point in merely repeating the details of the method. The constructive communication would be to point out if I am wrong when I say the method is just training a proxy classifier with transfer learning and thresholding predictions.
>
> We apologize for the confusion. Our contribution is indeed more than just training a proxy classifier with transfer learning and thresholding. Our proxy classifier is trained with uncertainty awareness and not transfer learning. The threshold is later applied on the uncertainty of predictions and not on the predictions themselves. The teacher network has the same architecture as the student network and the teacher’s weights are updated using the moving average of the student’s weights. We use Monte-Carlo Dropout over the teacher for uncertainty estimation and the consistency loss to enforce the classifier to focus on samples with low uncertainty.
>
> To the best of our knowledge, our proposed method is the first to draw connections between the uncertainty in sensitive attribute predictions and fairness. We have highlighted our contribution in the revised version of the draft.
>
> > Can authors explain the reason why no comparison to any of the methods in the literature of fairness without full access to sensitive attributes?.
>
> We have indeed performed comparisons with six other baselines. In the experiment section, we mentioned the comparison with other baselines addressing fairness issues in a similar setup and we referred the reader to the appendix for results. We performed comparisons with the following baselines:
>
> [1] Yan, S., Kao, H. T., & Ferrara, E. (2020, October). Fair class balancing: Enhancing model fairness without observing sensitive attributes. In Proceedings of the 29th ACM International Conference on Information & Knowledge Management
>
> [2] Zhao, T., Dai, E., Shu, K., & Wang, S. (2022, February). Towards fair classifiers without sensitive attributes: Exploring biases in related features. In Proceedings of the Fifteenth ACM International Conference on Web Search and Data Mining
>
> [3] Liang, Y., Chen, C., Tian, T., & Shu, K. (2023). Fair classification via domain adaptation: A dual adversarial learning approach. Frontiers in Big Data, 5, 129.
>
> [4] Lahoti, P., Beutel, A., Chen, J., Lee, K., Prost, F., Thain, N., ... & Chi, E. (2020). Fairness without demographics through adversarially reweighted learning. Advances in neural information processing systems
>
> [5] Hashimoto, T., Srivastava, M., Namkoong, H., & Liang, P. (2018, July). Fairness without demographics in repeated loss minimization. In International Conference on Machine Learning
>
> [6] Levy, D., Carmon, Y., Duchi, J. C., & Sidford, A. (2020). Large-scale methods for distributionally robust optimization. Advances in Neural Information Processing Systems
>
> These comparisons are presented in the appendix due to the page limit and the main objective of the proposed method, which is to demonstrate the utility of uncertainty awareness in the proxy classifier in improving fairness-accuracy tradeoffs in the downstream classifiers. The results presented in the appendix (Tables 3-6) also show the effectiveness of the method in providing better fairness-accuracy tradeoffs compared to the considered state-of-the-art baselines. Moreover, our evaluations are performed over a testing set with the true sensitive attributes, i.e., we report the true fairness violation. We do not aim to improve bias estimation when proxy attributes are used which is why some of the baselines [2] [3] [4]  proposed by the reviewer were not considered for comparison.
>
> We hope our response clarifies the reviewer’s concerns and we will be happy to provide further clarifications where needed.

---

> > ### Comment · Reviewer_PEk1 · 2023-11-20
> > **Response to the Rebuttal**
> >
> > I thank the authors for the rebuttal.
> >
> > Regarding technical novelty, unfortunately, the rebuttal does not seem convincing enough to me although the authors disagreed and mentioned it in the response to Reviewer EiNj. First, the authors stress that this is the first work that "draw connections between the uncertainty in sensitive attribute predictions and fairness." I think it depends on what is meant by "uncertainty." Technically speaking, when you do not have sensitive attributes and have to estimate them, any estimation would not be 100% certain and therefore one can argue any method that tries to predict the missing sensitive attribute has the flavor of uncertainty in it. If the word uncertain means in its literal sense, I am afraid this claim is too broad. Any method that tries to solve fairness problems with incomplete information and therefore has a probabilistic modeling of sensitive attributes can be considered as an uncertainty-based method, e.g. [1]. Hence it would be an overclaim. If the word uncertainty has a more technical meaning, I do not see any dedicated uncertainty method, e.g. conformal prediction, used in the paper.
> >
> > Overall, I still tend to think the method to be basic: training a better sensitive attribute proxy with some heuristic, and then threshold the confidence. The authors stressed in the rebuttal that thresholding is based on uncertainty estimation, but if I understand it correctly, what it does is nothing more than thresholding on the model's predicted confidence. Please correct me if I am wrong.
> >
> > Regarding the baseline comparison, I agree with Reviewer EiNj that the comparison to the baselines when the sensitive attribute is missing should be put into the main text because comparing to the methods that assume full sensitive attribute is not apple-to-apple and therefore meaningless. I also agree with Reviewer EiNj that "the comparisons involving Proxy-kNN and Proxy-DNN could be relegated to the appendix."
> >
> > In terms of Table 6 in the Appendix, it seems strange that the proposed method can beat all baselines with such a big margin: In DP, the best baseline is 0.02 while the proposed method is 0.003; in EOP numbers are 0.11 vs. 0.001; in EO the numbers are 0.129 vs. 0.007. It indicates the proposed method is over 20x better than the best baseline. I am suspicious that the method can be so effective. Please let me know if you have any convincing reasons.
> >
> > [1] Awasthi, Pranjal, et al. "Evaluating fairness of machine learning models under uncertain and incomplete information." Proceedings of the 2021 ACM Conference on Fairness, Accountability, and Transparency. 2021.

---

> > > ### Author Response · Authors · 2023-11-21
> > > **Reply to reviewer PEk1 feedback**
> > >
> > > We thank the reviewer for the response and for engaging in the discussion.
> > >
> > > Existing methods assume the sensitive attribute is uncertain due to various reasons such as noise or estimation (similarly to [1]). We quantify the uncertainty and provide evidence of its impact on the fairness-accuracy tradeoffs, while examples such as [1] tackle the issue differently (bounding the true fairness violation) and this makes our proposal novel while being simple. We used Monte-Carlo Dropout for uncertainty estimation along with a semi-supervised loss function to enforce the training of attribute classifier to account for the uncertainty in the dataset without sensitive attributes. Monte-Carlo Dropout is an approximation of the Bayesian Neural Networks and it can effectively capture various sources of uncertainties (Aleatoric and Epistemic uncertainty), which performed sufficiently well in our setup. We agree that this is not the only method, conformal prediction or even simple probabilities are also an option.
> > >
> > > Regarding Table 6, the performance gap is justified by the performances of the underlying fairness-enhancing technique used by our approach (Exponentiated Gradient [2]). Exponentiated Gradient offers an effective way to control fairness violations. For example, with the fairness violation set to 0, the method can derive a classifier that does not exceed the specified fairness violation, at least with some small margins, while minimizing the classification error. In Table 6, Exponentiated Gradient is the baseline method that uses the true sensitive attributes (Vanilla (with fairness)) with fairness violation set to 0, and it already provides a significant gap compared to other baselines on this dataset (DP: 0.008; EOP:  0.018, OED: 0.017). Applying our method, i.e., using Exponentiated Gradient as the fairness enhancing method with fairness violation to 0, but applying fairness constraints over samples with more certain sensitive attributes, improves the tradeoffs. These results follow our hypothesis and explain the gap compared to other baselines.
> > >
> > > Furthermore, the source code is provided for reproduction and the instructions to run each baseline are also provided ‘https://anonymous.4open.science/r/source-code-E86F/README.md’. We would appreciate if the reviewer could point out inconsistencies in the code that can support their claim on suspicious results.
> > >
> > > > Regarding the baseline comparison, I agree with Reviewer EiNj that the comparison to the baselines when the sensitive attribute is missing should be put into the main text because comparing to the methods that assume full [...]
> > >
> > > We moved the comparison with other baselines in the main text as recommended.
> > >
> > > We thank again the reviewer for engaging in the discussion and hope we have addressed all the concerns. We will be happy to provide further clarifications.
> > >
> > >
> > > [1] Awasthi, Pranjal, et al. "Evaluating fairness of machine learning models under uncertain and incomplete information." Proceedings of the 2021 ACM Conference on Fairness, Accountability, and Transparency. 2021.
> > >
> > > [2] Agarwal, A., Beygelzimer, A., Dudík, M., Langford, J., & Wallach, H. (2018, July). A reductions approach to fair classification. In International conference on machine learning (pp. 60-69). PMLR.

---

> > > > ### Comment · Reviewer_PEk1 · 2023-11-22
> > > > **Reply to the response**
> > > >
> > > > I thank the authors for replying and reorganizing the experimental section. The discussion will be taken into consideration.

---

> ### Author Response · Authors · 2023-11-23
> **Official Comment by Authors**
>
> Dear Reviewer PEk1,
>
> We Thank you for your time and for the constructive discussion, which has helped us to improve the manuscript. We have integrated all the suggestions you requested throughout our discussion and we will be happy to provide further clarifications and improvement where needed within the remaining rebuttal time. In case we have addressed all the concerns, we would like to request if you can adjust the score accordingly.

---

### Author Response · Authors · 2023-11-23
**General response**

We express our sincere gratitude to all the reviewers for their time, valuable feedback, and suggestions, which helped to significantly improve the manuscript throughout the discussion. We have incorporated all the suggestions requested and hope the current version of the manuscript addresses all your concerns.

**Main changes in the revision**

**Limitations**: We have added a limitation section where we discuss the issues collected from reviewer feedback, in particular the ethical concern in inferring sensitive information and the use of the true sensitive attributes for evaluation.


**Ethical concerns**: Following the feedback from reviewer EiNj, we have included an ethical statement concerning the ethical and legal issues of predicting sensitive information. We provided additional experiments recommended by the reviewer EiNj to explore alternative uses of our framework that would not require predicting sensitive information, which is more ethically acceptable.


**Alternative approach**:  Following reviewer EiNj feedback, we provided an alternative approach (OURS (uncertain)) where our framework is used to identify instances where the uncertainty of sensitive information is higher. These instances are then used to train a non-fairness-aware algorithm which yields fairness results without a high impact on the accuracy. Comparison with other baselines showed promising results across different datasets. We have added this approach to all the tables for comparison with other methods showing its performances against other baselines.
We also analyzed the impact of the uncertainty of the sensitive attributes in the data over the fairness of the downstream classifier, our results (Appendix D) show that the model gets fairer as the uncertainty on the sensitive attributes in the data increases while a slight drop in the accuracy is observed.


**Baselines**: As recommended by reviewer EiNj and PEk1 we have restructured the experiment section and moved the comparison with existing methods in the main text, and the comparison with other proxy classifiers (Proxy-KNN and Proxy-DNN) to the appendix.

**Additional Results:**  We have added a comparison with other baselines using a multi-layer perceptron as base classifiers.


**Related work**: Following reviewer EiNj recommendation, we have added privacy-preserving approaches in the related work section to provide additional perspective of the problem studied.


**Other changes**: We fixed the inconsistencies in the tables making the number of decimals uniform. We clarified how the uncertainty threshold can be tuned in practice.  We have emphasized in the main paper that the proposed framework does not perform transfer learning, although it uses a student-teacher structure.

  We thank the reviewers again for the engaging discussions and valuable feedback.

---

### Meta-Review · Area_Chair_JsUa · 2023-12-09

**Metareview:**

The paper explores achieving a more favorable fairness-accuracy tradeoff in scenarios where access to complete sensitive attributes is unavailable. The proposed method involves two steps: firstly, training a proxy classifier through student-teacher distillation to predict missing sensitive attributes, and secondly, applying confidence thresholding on predictions and utilizing instances with the least uncertain predictions. The outcome is the proxy sensitive information that can help effectively train the fairness-enhancing intervention. The method is evaluated on commonly used fairness benchmark datasets.

The proposed solution lacks novelty and the authors are encouraged to 1) better position their work and compare with existing line of results that seem to work on very similar problems and leverage similar techniques; and 2) to better justify the technical merits of their solution via providing theoretical insights.

In addition to the unclear technical merits, the paper, unfortunately, violates double blind policy: at line 3 of the anonymous link submitted by the authors, the LICENCE file, it states:
Copyright (c) 2023 [XXX]
where "[XXX]" seems to be the author's name.

**Justification For Why Not Higher Score:**

The paper is rejected for primarily two reasons: 1) unclear technical merits over existing large body of literature and 2) violation of double blind policy.

**Justification For Why Not Lower Score:**

N/A

---

### Decision · Program_Chairs · 2024-01-16

Reject